
# Comprehensive evaluations of diurnal NO₂ measurements during DISCOVER-AQ 2011: Effects of resolution dependent representation of NOₓ emissions

Jianfeng Li[1, a], Yuhang Wang[1*], Ruixiong Zhang[1], Charles Smeltzer[1], Andrew Weinheimer[2], Jay Herman[3], K. Folkert Boersma[4, 5], Edward A. Celarier[6, 7, b], Russell W. Long[8], James J. Szykman[8], Ruben Delgado[3], Anne M. Thompson[6], Travis N. Knepp[9, 10], Lok N. Lamsal[6], Scott J. Janz[6], Matthew G. Kowalewski[6], Xiong Liu[11], Caroline R. Nowlan[11]

[1]School of Earth and Atmospheric Sciences, Georgia Institute of Technology, Atlanta, Georgia, USA
[2]National Center for Atmospheric Research, Boulder, Colorado, USA
[3]University of Maryland Baltimore County JCET, Baltimore, Maryland, USA
[4]Royal Netherlands Meteorological Institute, De Bilt, the Netherlands
[5]Wageningen University, Meteorology and Air Quality Group, Wageningen, the Netherlands
[6]NASA Goddard Space Flight Center, Greenbelt, Maryland, USA
[7]Universities Space Research Association, Columbia, Maryland, USA
[8]National Exposure Research Laboratory, Office of Research and Development, U.S. Environmental Protection Agency, Research Triangle Park, NC, USA
[9]NASA Langley Research Center, Virginia, USA
[10]Science Systems and Applications, Inc., Hampton, Virginia, USA
[11]Harvard-Smithsonian Center for Astrophysics, Cambridge, Massachusetts, USA

[a]now at: Atmospheric Sciences and Global Change Division, Pacific Northwest National Laboratory, Richland, Washington, USA
[b]now at: Digital Spec, Tyson's Corner, VA, USA

*Correspondence to* Yuhang Wang (yuhang.wang@eas.gatech.edu)



# Abstract

Nitrogen oxides ($NO_x = NO + NO_2$) play a crucial role in the formation of ozone and secondary inorganic and organic aerosols, thus affecting human health, global radiation budget, and climate. The diurnal and spatial variations of $NO_2$ are functions of emissions, advection, deposition, vertical mixing, and chemistry. Their observations, therefore, provide useful constraints in our understanding of these factors. We employ a Regional chEmical and trAnsport model (REAM) to analyze the observed temporal (diurnal cycles) and spatial distributions of $NO_2$ concentrations and tropospheric vertical column densities (TVCDs) using aircraft in situ measurements, surface EPA Air Quality System (AQS) observations, as well as the measurements of TVCDs by satellite instruments (OMI: the Ozone Monitoring Instrument; and GOME-2A: Global Ozone Monitoring Experiment – 2A), ground-based Pandora, and the Airborne Compact Atmospheric Mapper (ACAM) instrument, in July 2011 during the DISCOVER-AQ campaign over the Baltimore-Washington region. The model simulations at 36- and 4-km resolutions are in reasonably good agreement with the temporospatial $NO_2$ observations in the daytime. However, nighttime mixing in the model needs to be enhanced to reproduce the observed $NO_2$ diurnal cycle in the model. Another discrepancy is that Pandora measured $NO_2$ TVCDs show much less variation in the late afternoon than simulated in the model. Relative to the 36-km model simulations, the 4-km model results show larger biases compared to the observations due largely to the larger spatial variations of $NO_2$ in the model when the spatial resolution is increased from 36 to 4 km, although the biases are often comparable to the ranges of the observations. The high-resolution aircraft ACAM observations show a more dispersed distribution of $NO_2$ vertical column densities (VCDs) and lower VCDs in urban regions than 4-km model simulations, reflecting likely the spatial distribution bias of $NO_x$ emissions in the National Emissions Inventory (NEI) 2011 at high resolution.





# 1 Introduction

Nitrogen oxides ($NO_x = NO + NO_2$) are among the most important trace gases in the atmosphere due to their crucial role in the formation of ozone ($O_3$), secondary aerosols, and their role in the chemical transformation of other atmospheric species, such as carbon monoxide (CO) and volatile organic compounds (VOCs) (Cheng et al., 2017; Cheng et al., 2018; Fisher et al., 2016; Li et al., 2019; Liu et al., 2012; Ng et al., 2017; Peng et al., 2016; Zhang and Wang, 2016). $NO_x$ is emitted by both anthropogenic activities and natural sources. Anthropogenic sources account for about 77% of the global $NO_x$ emissions, and fossil fuel combustion and industrial processes are the primary anthropogenic sources, which contribute to about 75% of the anthropogenic emissions (Seinfeld and Pandis, 2016). Other important anthropogenic sources include agriculture and biomass and biofuel burning. Soils and lightning are two major natural sources. Most $NO_x$ is emitted as NO, which is then oxidized to $NO_2$ by oxidants, such as $O_3$, the hydroperoxyl radical ($HO_2$), and organic peroxy radicals ($RO_2$).

The diurnal variations of $NO_2$ controlled by physical and chemical processes reflect the temporal patterns of these underlying controlling factors, such as $NO_x$ emissions, chemistry, deposition, advection, diffusion, and convection. Therefore, the observations of $NO_2$ diurnal cycles can be used to evaluate our understanding of $NO_x$ related emission, chemistry, and physical processes (Frey et al., 2013; Jones et al., 2000; Judd et al., 2018). For example, Brown et al. (2004) analyzed the diurnal patterns of surface NO, $NO_2$, $NO_3$, $N_2O_5$, $HNO_3$, OH, and $O_3$ concentrations along the East Coast of the United States (U.S.) during the New England Air Quality Study (NEAQS) campaign in the summer of 2002 and found that the predominant nighttime sink of $NO_x$ through the hydrolysis of $N_2O_5$ had an efficiency on par with daytime photochemical loss over the ocean surface off the New England coast. Van Stratum et al. (2012) investigated the contribution of boundary layer dynamics to chemistry evolution during the DOMINO (Diel Oxidant Mechanisms in relation to Nitrogen Oxides) campaign in 2008 in Spain and found that entrainment and boundary layer growth in daytime influenced mixed-layer NO and $NO_2$





diurnal cycles on the same order of chemical transformations. David and Nair (2011) found that the diurnal
patterns of surface NO, $NO_2$, and $O_3$ concentrations at a tropical coastal station in India from November 2007 to
May 2009 were closely associated with sea breeze and land breeze which affected the availability of $NO_x$ through
transport. They also thought that monsoon-associated synoptic wind patterns could strongly influence the
magnitudes of NO, $NO_2$, and $O_3$ diurnal cycles. The monsoon effect on surface NO, $NO_2$, and $O_3$ diurnal cycles
was also observed in China by Tu et al. (2007) on the basis of continuous measurements of NO, $NO_2$, and $O_3$ at
an urban site in Nanjing from January 2000 – February 2003.
In addition to surface $NO_2$ diurnal cycles, the daily variations of $NO_2$ vertical column densities (VCDs) were
also investigated in previous studies. For example, Boersma et al. (2008) compared $NO_2$ tropospheric VCDs
(TVCDs) retrieved from OMI (the Ozone Monitoring Instrument) and SCIAMACHY (SCanning Imaging
Absorption SpectroMeter for Atmospheric CHartography) in August 2006 around the world. They found that the
diurnal patterns of different types of $NO_x$ emissions could strongly affect the $NO_2$ TVCD variations between
OMI and SCIAMACHY and that intense afternoon fire activity resulted in an increase of $NO_2$ TVCDs from
10:00 LT (local time) to 13:30 LT over tropical biomass burning regions. Boersma et al. (2009) further
investigated the $NO_2$ TVCD change from SCIAMACHY to OMI in different seasons of 2006 in Israeli cities and
found that there was a slight increase of $NO_2$ TVCDs from SCIAMACHY to OMI in winter due to increased $NO_x$
emissions from 10:00 LT to 13:30 LT and a sufficiently weak photochemical sink and that the TVCDs from OMI
were lower than SCIAMACHY in summer due to a strong photochemical sink of $NO_x$.
All these above researches, however, exploited only $NO_2$ surface or satellite VCD measurements. Knepp et
al. (2015) related the daytime variations of $NO_2$ TVCD measurements by ground-based Pandora instruments to
the variations of coincident $NO_2$ surface concentrations using a planetary boundary layer height (PBLH) factor
over the periods July 2011 – October 2011 at the NASA Langley Research Center in Hampton, Virginia and July





2011 at Padonia and Edgewood sites in Maryland for the DISCOVER-AQ experiment, showing the importance
of boundary-layer vertical mixing on $NO_2$ vertical distributions and the ability of $NO_2$ VCD measurements to
infer hourly boundary-layer $NO_2$ variations. DISCOVER-AQ, the Deriving Information on Surface conditions
from Column and Vertically Resolved Observations Relevant to Air Quality experiment (https://discover-
aq.larc.nasa.gov/), was designed to better understand the relationship between boundary-layer pollutants and
satellite observations (Flynn et al., 2014; Reed et al., 2015). Figure S1 shows the sampling locations of the
summer DISCOVER-AQ 2011 campaign in the Baltimore-Washington metropolitan region. In this campaign, the
NASA P-3B aircraft flew spirals over six air quality monitoring sites (Aldino - rural/suburban, Edgewood -
coastal/urban, Beltsville - suburban, Essex - coastal/urban, Fairhill - rural, and Padonia - suburban) (Table S1)
and the Chesapeake Bay (Cheng et al., 2017; Lamsal et al., 2014), and measured 244 $NO_2$ profiles in 14 flight
days in July (Zhang et al., 2016). During the same period, the NASA UC-12 aircraft flew across the Baltimore-
Washington region at an altitude about 8 km above sea level (ASL), using the Airborne Compact Atmospheric
Mapper (ACAM) to map the distributions of $NO_2$ VCDs below the aircraft (Lamsal et al., 2017). Furthermore,
ground-based instruments were deployed to measure $NO_2$ surface concentrations, $NO_2$ VCDs, and other physical
properties of the atmosphere (Anderson et al., 2014; Reed et al., 2015; Sawamura et al., 2014). Satellite OMI and
GOME-2A (Global Ozone Monitoring Experiment – 2A) instruments provided $NO_2$ TVCD measurements over
the campaign region at 13:30 and 9:30 LT, respectively. These concurrent measurements of $NO_2$ VCDs, surface
$NO_2$, and vertically resolved distributions of $NO_2$ during the DISCOVER-AQ 2011 campaign, therefore, provide
a comprehensive dataset to evaluate $NO_2$ diurnal and spatial variabilities and processes affecting $NO_2$
concentrations.

Section 2 describes the measurement datasets in detail.  The Regional chEmistry and trAnsport Model

(REAM), also described in section 2, is applied to simulate the $NO_2$ observations during the DISCOVER-AQ
campaign in July 2011. The evaluations of the simulated diurnal cycles of surface $NO_2$ concentrations, $NO_2$





vertical profiles, and $NO_2$ TVCDs are discussed in section 3 through comparisons with observations. In section 3,
we also investigate the differences between $NO_2$ diurnal cycles on weekdays and weekends and their implications
for $NO_x$ emission characteristics. To corroborate our evaluation of $NO_x$ emissions based on $NO_2$ diurnal cycles,
we further compare observed $NO_y$ (reactive nitrogen compounds) concentrations with REAM simulation results
in section 3. Moreover, we assess the resolution dependence of REAM simulation results in light of the
observations and discuss the potential biases of $NO_x$ emissions at high resolution by comparing the 4-km REAM
simulation results with high-resolution ACAM $NO_2$ VCDs. Finally, we summarize the study in section 4.

## 2 Datasets and model description

2.1 REAM
REAM has been widely applied in many studies (Cheng et al., 2017; Choi et al., 2008; Li et al., 2019; Zhang
et al., 2018; Zhang et al., 2016; Zhao et al., 2009). The model has a horizontal resolution of 36 km and 30 vertical
layers in the troposphere. Meteorology fields are from a Weather Research and Forecasting (WRF, version 3.6)
model simulation with a horizontal resolution of 36 km. We summarize the physics parameterization schemes of
the WRF simulation in Table S2. The WRF simulation is initialized and constrained by the NCEP coupled
forecast system model version 2 (CFSv2) products (http://rda.ucar.edu/datasets/ds094.0/) (Saha et al., 2011). The
chemistry mechanism in REAM is based on GEOS-Chem v11.01 with updated aerosol uptake of isoprene nitrates
(Fisher et al., 2016) and revised treatment of wet scavenging processes (Luo et al., 2019). A $2° \times 2.5°$ GEOS-
Chem simulation provides the chemical boundary and initial conditions.
Biogenic VOC emissions in REAM are from MEGAN v2.10 (Guenther et al., 2012). Anthropogenic
emissions on weekdays are from the National Emission Inventory 2011 (NEI2011) (EPA, 2014) from the Pacific
Northwest National Laboratory (PNNL), which has an initial resolution of 4 km and is re-gridded to REAM 36-





km grid cells (Figure S2). Weekday emission diurnal profiles are from NEI2011. The weekday to weekend
emission ratios and weekend emission diurnal profiles are based on previous studies (Beirle et al., 2003; Boersma
et al., 2009; Choi et al., 2012; de Foy, 2018; DenBleyker et al., 2012; Herman et al., 2009; Judd et al., 2018;
Kaynak et al., 2009; Kim et al., 2016). These studies suggested that weekend $NO_x$ emissions were 20% - 50%
lower than weekday emissions, and the weekend $NO_x$ emission diurnal cycles were different from weekdays;
therefore, we specify a weekend to weekday $NO_x$ emission ratio of 2/3 in this study. The resulting diurnal
variations of weekday and weekend $NO_x$ emissions over the DISCOVER-AQ 2011 region are shown in Figure 1.
The diurnal emission variation is lower on weekends than on weekdays.
To understand the effects of model resolutions on the temporospatial distributions of $NO_2$, we also conduct a
REAM simulation with a horizontal resolution of 4 km during the DISCOVER-AQ campaign. A 36-km REAM
simulation (discussed in section 3.1) provides the chemical initial and hourly boundary conditions. Meteorology
fields are from a nested WRF simulation (36 km, 12 km, 4 km) with cumulus parameterization turned off in the
4-km domain (Table S2). Figure S1 shows a comparison of the 4-km and 36-km REAM grid cells with
DISCOVER-AQ observations, and Figure S2 shows a comparison of $NO_x$ emission distributions between the 4-
km and 36-km REAM simulations. The comparison of $NO_x$ emission diurnal variations over the DISCOVER-AQ
2011 region between the 4-km and 36-km REAM is shown in Figure 1.
We evaluate the performances of the 36-km and nested 4-km WRF simulations by comparing temperature
and wind from the P-3B spirals (Figure S1) and precipitation from the NCEP (National Centers for
Environmental Prediction) Stage IV precipitation dataset with those coincident WRF simulation results in July
2011. Generally, P-3B spirals range from ~400 m to ~3.63 km in height above the ground level (AGL). As shown
in Figure S3, both the 36-km and nested 4-km WRF simulations predict temperature well with $R^2 = 0.94$ and $R^2 =$
0.98, respectively. Both WRF simulations show good agreement with P-3B measurements in U-wind (36-km: $R^2$





= 0.62; 4-km: $R^2$ = 0.71), V-wind (36-km: $R^2$ = 0.75; 4-km: $R^2$ = 0.74), wind speed (36-km: $R^2$ = 0.52; 4-km: $R^2$
= 0.64), and wind direction (36-km: $R^2$ = 0.40; 4-km: $R^2$ = 0.50) (Figures S3 and S4). The evaluations above
suggest that WRF simulated wind fields are good and comparable at 4-km and 36-km resolutions; they are not the
reasons for the differences of the 4-km and 36-km simulations of trace gases by REAM, which is driven by WRF
meteorological fields.
The NCEP Stage IV precipitation dataset provides hourly precipitation across the contiguous United States
(CONUS) with a resolution of ~4 km based on the merging of rain gauge data and radar observations (Lin and
Mitchell, 2005; Nelson et al., 2016). The Stage IV dataset is useful for evaluating model simulations, satellite
precipitation estimates, and radar precipitation estimates (Davis et al., 2006; Gourley et al., 2011; Kalinga and
Gan, 2010; Lopez, 2011; Yuan et al., 2008). We obtain the Stage IV precipitation data in July 2011 from the
NCAR/UCAR Research Data Archive (https://rda.ucar.edu/datasets/ds507.5/). As shown in Figures S5 and S6,
generally, both the 36-km and nested 4-km WRF simulations predict much less precipitation (in precipitation
amount and duration) compared to Stage-IV in July 2011 around the DISCOVER-AQ campaign region,
especially for the nested 4-km WRF simulation. We find that large-scale precipitation amounts are much less
compared to convective precipitation in most regions in the 36-km WRF simulation (Figure S7) during the
simulation period, which is contradictory to Li et al. (2020) showing non-convective precipitation accounting for
25% – 40% of the total precipitation. At 4-km resolution, convective and non-convective precipitations are not
separated because convection is explicitly resolved. The model low bias is large (Figure S6). The underestimation
of precipitation in our WRF simulations may lead to high biases of soluble species in REAM, such as $HNO_3$, due
to underestimated wet scavenging.





2.2 NO$_2$ TVCD measurements by OMI and GOME-2A

The OMI instrument onboard the sun-synchronous NASA EOS Aura satellite with an equator-crossing time

of around 13:30 LT was developed by the Finnish Meteorological Institute and the Netherlands Agency for
Aerospace Programs to measure solar backscattering radiation in the visible and ultraviolet bands (Levelt et al.,
2006; Russell et al., 2012). The radiance measurements are used to derive trace gas concentrations in the
atmosphere, such as O$_3$, NO$_2$, HCHO, and SO$_2$ (Levelt et al., 2006). OMI has a nadir resolution of 13 km × 24 km
and provides daily global coverage (Levelt et al., 2006).

Two widely-used archives of OMI NO$_2$ VCD products are available, NASA OMNO2 (v4.0)

(https://disc.gsfc.nasa.gov/datasets/OMNO2_003/summary) and KNMI DOMINO (v2.0)
(http://www.temis.nl/airpollution/no2.html). Although both use Differential Optical Absorption Spectroscopy
(DOAS) algorithms to derive NO$_2$ slant column densities, they have differences in spectral fitting, stratospheric
and tropospheric NO$_2$ slant column density (SCD) separation, a priori NO$_2$ vertical profiles, and air mass factor
(AMF) calculation, etc. (Boersma et al., 2011; Bucsela et al., 2013; Chance, 2002; Krotkov et al., 2017; Lamsal
et al., 2020; Marchenko et al., 2015; Oetjen et al., 2013; van der A et al., 2010; Van Geffen et al., 2015). Both
OMNO2 and DOMINO have been extensively evaluated with field measurements and models (Boersma et al.,
2009; Boersma et al., 2011; Choi et al., 2020; Hains et al., 2010; Huijnen et al., 2010; Ionov et al., 2008; Irie et
al., 2008; Lamsal et al., 2014; Lamsal et al., 2020; Oetjen et al., 2013). The estimated uncertainty of DOMINO
TVCD product is $1.0 \times 10^{15}$ molecules cm$^{-2}$ + 25% (Boersma et al., 2011), while the uncertainty of OMNO2
TVCD product ranges from ~30% under clear-sky conditions to ~60% under cloudy conditions (Lamsal et al.,
2014; Oetjen et al., 2013; Tong et al., 2015). In order to reduce uncertainties in this study, we only use TVCD
data with effective cloud fractions < 0.2. The data affected by row anomaly are excluded (Boersma et al., 2018;
Zhang et al., 2018).



For AMF calculation, DOMINO used daily TM4 model results with a resolution of $3° \times 2°$ as a priori $NO_2$
vertical profiles (Boersma et al., 2007; Boersma et al., 2011), while OMNO2 v4.0 used monthly mean values
from the Global Modeling Initiative (GMI) model with a resolution of $1° \times 1.25°$. The relatively coarse horizontal
resolution of the a priori $NO_2$ profiles in the retrievals can introduce uncertainties in the spatial and temporal
characteristics of $NO_2$ TVCDs at satellite pixel scales. For comparison purposes, we also use 36-km REAM
simulation results as the a priori $NO_2$ profiles to compute the AMFs and $NO_2$ TVCDs with the DOMINO
algorithm.
The GOME-2 instrument onboard the polar-orbiting MetOp-A satellite (now referred to as GOME-2A) is an
improved version of GOME-1 launched in 1995 and has an overpass time of 9:30 LT and a spatial resolution of
$80 \times 40$ km$^2$ (Munro et al., 2006; Peters et al., 2012). GOME-2A measures backscattered solar radiation in the
range from 240 nm to 790 nm, which is used for VCD retrievals of trace gases, such as $O_3$, $NO_2$, BrO, and $SO_2$
(Munro et al., 2006). We use the KNMI TM4NO2A v2.3 GOME-2A $NO_2$ VCD product archived on
http://www.temis.nl/airpollution/no2col/no2colgome2_v2.php (Boersma et al., 2007; Boersma et al., 2011).
GOME-2A derived $NO_2$ VCDs have been validated with SCIAMACHY and MAX-DOAS measurements (Irie et
al., 2012; Peters et al., 2012; Richter et al., 2011). As in the case of OMI, we also recalculate the AMF values and
GOME-2A TVCDs using the daily 36-km REAM $NO_2$ profiles (9:00 LT – 10:00 LT).
2.3 Pandora ground-based $NO_2$ VCD measurements
Pandora is a small direct sun spectrometer, which measures sun and sky radiance from 270 to 530 nm with a
0.5 nm resolution and a $1.6°$ field of view (FOV) for the retrieval of the total VCDs of $NO_2$ with a precision of
about $2.7 \times 10^{14}$ molecules/cm$^2$ and a nominal accuracy of $2.7 \times 10^{15}$ molecules cm$^{-2}$ under clear-sky conditions
(Herman et al., 2009; Lamsal et al., 2014). There were 12 Pandora sites operating in the DISCOVER-AQ
campaign (Figure S1). Six of them are the same as the P-3B aircraft spiral locations (Aldino, Edgewood,


Beltsville, Essex, Fairhill, and Padonia) (Table S1 and Figure S1). The other six sites are Naval Academy
(Annapolis Maryland) (USNA – ocean), University of Maryland College Park (UMCP – urban), University of
Maryland Baltimore County (UMBC – urban), Smithsonian Environmental Research Center (SERC –
rural/coastal), Oldtown in Baltimore (Oldtown – urban), and Goddard Space Flight Center (GSFC –
urban/suburban) (Table S1 and Figure S1). In this study, we exclude the USNA site as its measurements were
conducted on a ship ("Pandora(w)" in Figure S1), and there were no other surface observations in the
corresponding REAM grid cell. Including the data from the USNA site has a negligible effect on the comparisons
of observed and simulated $NO_2$ TVCDs. In our analysis, we ignore Pandora measurements with solar zenith
angles (SZA) > 80° (Figure S8) and exclude the data when fewer than three valid measurements are available
within an hour to reduce the uncertainties of the hourly averages due to the significant variations of Pandora
observations (Figure S9).
Since Pandora measures total $NO_2$ VCDs, we need to subtract stratosphere $NO_2$ VCDs from the total VCDs
to compute TVCDs. As shown in Figure S10, stratosphere $NO_2$ VCDs show a clear diurnal cycle with an increase
during daytime due in part to the photolysis of reactive nitrogen reservoirs such as $N_2O_5$ and $HNO_3$ (Brohede et
al., 2007; Dirksen et al., 2011; Peters et al., 2012; Sen et al., 1998; Spinei et al., 2014), which is consistent with
the significant increase of stratospheric $NO_2$ VCDs from GOME-2A to OMI. In this study, we use the GMI
model simulated stratospheric $NO_2$ VCDs in Figure S10 to calculate the Pandora $NO_2$ TVCDs. The small
discrepancies between the GMI stratospheric $NO_2$ VCDs and satellite products do not change the pattern of
Pandora $NO_2$ TVCD diurnal variations or affect the conclusions in this study.
2.4 ACAM $NO_2$ VCD measurements
The ACAM instrument onboard the UC-12 aircraft consists of two thermally spectrometers in the
ultraviolet/visible/near-infrared range. The spectrometer in the ultraviolet/visible band (304 nm – 520 nm) with a



resolution of 0.8 nm and a sampling of 0.105 nm can be used to detect $NO_2$ in the atmosphere. The native ground
resolution of UC-12 ACAM $NO_2$ measurements is 0.5 km $\times$ 0.75 km at a flight altitude of about 8 km ASL and a
nominal ground speed of 100 m s$^{-1}$ during the DISCOVER-AQ 2011 campaign (Lamsal et al., 2017), thus
providing high-resolution $NO_2$ VCDs below the aircraft.
In this study, we mainly use the ACAM $NO_2$ VCD product described by Lamsal et al. (2017), which applied
a pair-average co-adding scheme to produce $NO_2$ VCDs at a ground resolution of about 1.5 km (cross-track) $\times$
1.1 km (along-track) to reduce noise impacts. In their retrieval of ACAM $NO_2$ VCDs, they first used the DOAS
fitting method to generate differential $NO_2$ SCDs relative to the SCDs at an unpolluted reference location. Then
they computed above/below-aircraft AMFs at both sampling and reference locations based on the vector
linearized discrete ordinate radiative transfer code (VLIDORT) (Spurr, 2008). In the computation of AMFs, the a
priori $NO_2$ vertical profiles were from a combination of a high-resolution (4-km) CMAQ (the Community
Multiscale Air Quality Modeling System) model outputs in the boundary layer and a GMI simulation (2° $\times$ 2.5°)
results elsewhere in the atmosphere. Finally, the below-aircraft $NO_2$ VCDs at the sampling locations were
generated by dividing below-aircraft $NO_2$ SCDs at the sampling locations by the corresponding below-aircraft
AMFs. The below-aircraft $NO_2$ SCDs were the differences between the total and above-aircraft $NO_2$ SCDs. The
total $NO_2$ SCDs were the sum of DOAS fitting generated differential $NO_2$ SCDs and $NO_2$ SCDs at the reference
location, and the above-aircraft $NO_2$ SCDs were derived based on above-aircraft AMFs, GMI $NO_2$ profiles, and
OMNO2 stratospheric $NO_2$ VCDs (Lamsal et al., 2017). The ACAM $NO_2$ VCD product had been evaluated via
comparisons with other independent observations during the DISCOVER-AQ 2011 campaign, such as P-3B
aircraft, Pandora, and OMNO2, and the uncertainty of individual below-aircraft $NO_2$ VCD is about 30% (Lamsal
et al., 2017). To keep the consistency of ACAM $NO_2$ VCDs, we exclude $NO_2$ VCDs measured at altitudes < 8 km
ASL, which accounts for about 6.8% of the total available ACAM $NO_2$ VCD data. We regrid the 1.5 km $\times$ 1.1
km ACAM $NO_2$ VCDs to the 4-km REAM grid cells (Figure S1), which are then used to evaluate the distribution



of NO$_2$ VCDs in the 4-km REAM simulation. As a supplement in section 3.6, we also assess the 4-km REAM
simulation by using the UC-12 ACAM NO$_2$ VCDs produced by the Smithsonian Astrophysical Observatory
(SAO) algorithms, archived on https://www-air.larc.nasa.gov/cgi-bin/ArcView/discover-aq.dc-
2011?UC12=1#LIU.XIONG/ (Liu et al., 2015a; Liu et al., 2015b). This product is an early version of the SAO
algorithm used to produce the Geostationary Trace gas and Aerosol Sensor Optimization (GeoTASO) and the
GEOstationary Coastal and Air Pollution Events (GEO-CAPE) Airborne Simulator (GCAS) airborne
observations in later airborne campaigns (Nowlan et al., 2016; Nowlan et al., 2018).
2.5 Surface NO$_2$ and O$_3$ measurements

The measurement of NO$_x$ is based on the chemiluminescence of electronically excited NO$_2^*$, produced from

the reaction of NO with O$_3$, and the strength of the chemiluminescence from the decay of NO$_2^*$ to NO$_2$ is
proportional to the number of NO molecules present (Reed et al., 2016). NO$_2$ concentrations can be measured
with this method by converting NO$_2$ to NO first through catalytic reactions (typically on the surface of heated
molybdenum oxide (MoO$_x$) substrate) or photolytic processes (Lamsal et al., 2015; Reed et al., 2016). However,
for the catalytic method, reactive nitrogen compounds other than NO$_x$ (NO$_z$), such as HNO$_3$, peroxyacetyl nitrate
(PAN), and other organic nitrates, can also be reduced to NO on the heated surface, thus causing an
overestimation of NO$_2$. The magnitude of the overestimation depends on the concentrations and the reduction
efficiencies of interference species, both of which are uncertain. The photolytic approach, which employs
broadband photolysis of ambient NO$_2$, offers more accurate NO$_2$ measurements (Lamsal et al., 2015).

There were 11 NO$_x$ monitoring sites operating in the DISCOVER-AQ region during the campaign (Figure

S1), including those from the EPA Air Quality System (AQS) monitoring network and those deployed for the
DISCOVER-AQ campaign. Nine of them measured NO$_2$ concentrations by a catalytic converter. The other two
sites (Edgewood and Padonia) had NO$_2$ measurements from both catalytic and photolytic methods. Different





stationary catalytic instruments were used during the campaign: Thermo Electron 42C-Y $NO_y$ analyzer, Thermo
Model 42C $NO_x$ analyzer, Thermo Model 42I-Y $NO_y$ analyzer, and Ecotech Model 9843/9841 T-$NO_y$ analyzers.
In addition, a mobile platform — NATIVE (Nittany Atmospheric Trailer and Integrated Validation Experiment)
with a Thermo Electron 42C-Y $NO_y$ analyzer installed, was also deployed in the Edgewood site. The photolytic
measurements of $NO_2$ in Edgewood and Padonia were from Teledyne API model 200eup photolytic $NO_x$
analyzers. We scale catalytic $NO_2$ measurements using the diurnal ratios of $NO_2$ photolytic measurements to $NO_2$
from the corresponding catalytic analyzers (Figure 2). Figure 2 shows the lowest photolytic/catalytic ratios in the
afternoon, which reflects the production of nitrates and other reactive nitrogen compounds from $NO_x$ in the
daytime. When photolytic measurements are available, we only use the photolytic observations in this study;
otherwise, we use the scaled catalytic measurements.
Nineteen surface $O_3$ monitoring sites were operating in the DISCOVER-AQ region during the campaign
(Figure S1). They measured $O_3$ concentrations by using a Federal Equivalent Method (FEM) based on the UV
absorption of $O_3$ (https://www.arb.ca.gov/aaqm/qa/qa-manual/vol4/chapter6o3.pdf) with an uncertainty of 5 ppb.
2.6 Aircraft measurements of $NO_2$ vertical profiles
In this study, we mainly use the $NO_2$ concentrations measured by the National Center for Atmospheric
Research (NCAR) 4-channel chemiluminescence instrument (P-CL) onboard the P-3B aircraft for the evaluation
of REAM simulated $NO_2$ vertical profiles. The instrument has a $NO_2$ measurement uncertainty of 10% – 15% and
a 1-second, 1-sigma detection limit of 30 pptv.
$NO_2$ measurements from aircraft spirals provide us with $NO_2$ vertical profiles. Figure S1 shows the locations
of the aircraft spirals during the DISCOVER-AQ campaign, except for the Chesapeake Bay spirals over the
ocean. There were only six spirals available over the Chesapeake Bay, which have ignorable impacts on the



following analyses. Therefore, we do not use them in this study. The rest 238 spirals in the daytime for July 2011
are used to compute the average profiles of $NO_2$ for the six inland sites (Figure S1).

The aircraft measurements were generally sampled from about a height of 400 m AGL in the boundary layer

to 3.63 km AGL in the free troposphere. We bin these measurements to REAM vertical levels. In order to make
up the missing observations between the surface and 400 m, we apply quadratic polynomial regressions by using
aircraft data below 1 km and coincident $NO_2$ surface measurements.

In addition to using $NO_2$ concentrations from the NCAR 4-channel instrument to evaluate REAM simulated

$NO_2$ vertical profiles, we also use P-3B NO, $NO_2$, and $NO_y$ concentrations measured by the NCAR 4-channel
instrument and $NO_2$, total peroxyacyl nitrates ($\sum$PNs), total alkyl nitrates ($\sum$ANs) (include alkyl nitrates and
hydroxyalkyl nitrates), and $HNO_3$ concentrations measured by the thermal dissociation-laser induced
fluorescence (TD-LIF) technique (Day et al., 2002; Thornton et al., 2000; Wooldridge et al., 2010) to evaluate the
concentrations of $NO_y$ from REAM (Table 1). All these P-3B measurements are vertically binned to REAM grid
cells for comparisons with REAM results. In addition, below the P-3B spirals, four $NO_y$ observation sites at
Padonia, Edgewood, Beltsville, and Aldino were operating to provide continuous hourly $NO_y$ surface
concentrations during the campaign, which we also use to evaluate REAM simulated $NO_y$ surface concentrations
in this study. We summarize the information of available observations at the 11 inland Pandora sites in Table S1.



# 3 Results and discussion

3.1 Effect of boundary layer vertical mixing on the diurnal variations of surface $NO_2$ concentrations

*3.1.1 36-km model simulation in comparison to the surface observations*

Figures 3a and 3b show the observed and 36-km REAM simulated diurnal cycles of surface $NO_2$ and $O_3$ concentrations on weekdays in July 2011 in the DISCOVER-AQ region. REAM with WRF simulated vertical diffusion coefficient ($k_{zz}$) values significantly overestimates $NO_2$ concentrations and underestimates $O_3$ concentrations at night, although it captures the patterns of the diurnal cycles of surface $NO_2$ and $O_3$: an $O_3$ peak and a $NO_2$ minimum around noontime. At night, the reaction of $O_3 + NO \rightarrow O_2 + NO_2$ produces $NO_2$ but removes $O_3$. Since most $NO_x$ emissions are in the form of NO, the model biases of low $O_3$ and high $NO_2$ occur at the same time. Since there are no significant chemical sources of $O_3$ at night, mixing of $O_3$ rich air above the surface is the main source of $O_3$ supply near the surface. Therefore, the nighttime model biases with WRF simulated $k_{zz}$ data in Figure 3 indicate that vertical mixing may be underestimated at night.

During the DISCOVER-AQ campaign, WRF simulated vertical wind velocities are very low at night and have little impact on vertical mixing (Figure S11a). The nighttime vertical mixing is mainly attributed to turbulent mixing. In the Yonsei University (YSU) planetary boundary layer (PBL) scheme (Shin and Hong, 2011) used by our WRF simulations (Table S2), boundary layer $k_{zz}$ is correlated to PBLH. However, Breuer et al. (2014) and Hu et al. (2012) found that the YSU scheme underestimated nighttime PBLHs in WRF, which is consistent with Figure 4 showing that YSU $k_{zz}$-determined PBLHs are significantly lower than lidar observations in the late afternoon and at night at the UMBC site during the DISCOVER-AQ campaign. The lidar mixing depth data were derived from the Elastic Lidar Facility (ELF) attenuated backscatter signals by using the covariance wavelet transform method and had been validated against radiosonde measurements, Radar wind profiler





observations, and Sigma Space mini-micropulse lidar data (Compton et al., 2013). To improve nighttime PBLHs
and vertical mixing in REAM, we increase REAM $k_{zz}$ below 500 m during 18:00 – 5:00 LT to 5 m s$^{-2}$ if the WRF
computed $k_{zz} < 5$ m s$^{-2}$, which significantly increases the PBLHs at night (Figure 4), leading to the decreases of
simulated surface $NO_2$ and the increases of surface $O_3$ concentrations at night (Figure 3). The assigned value of 5
m s$^{-2}$ is arbitrary. Changing this value to 2 or 10 m s$^{-2}$ can also alleviate the biases of model simulated nighttime
surface $NO_2$ and $O_3$ concentrations. An alternative solution to correct the model nighttime simulation biases is to
reduce $NO_x$ emissions by 50-67%, but we cannot find good reasons to justify this level of $NO_x$ emission
reduction only at night.

The updated REAM simulation of surface $NO_2$ diurnal pattern in Figure 3a is in good agreement with

previous studies (Anderson et al., 2014; David and Nair, 2011; Gaur et al., 2014; Reddy et al., 2012). Daytime
surface $NO_2$ concentrations are much lower compared to nighttime, and $NO_2$ concentrations reach a minimum
around noontime. As shown in Figure S12, under the influence of vertical turbulent mixing, the surface-layer
$NO_x$ emission diurnal pattern is similar to the surface $NO_2$ diurnal cycle in Figure 3a, emphasizing the importance
of turbulent mixing on modulating surface $NO_2$ diurnal variations. The highest boundary layer (Figure 4) due to
solar radiation leads to the lowest surface-layer $NO_x$ emissions (Figure S12) and, therefore, the smallest surface
$NO_2$ concentrations occur around noontime (Figure 3a). Transport, which is mainly attributed to advection and
turbulent mixing, is another critical factor affecting surface $NO_2$ diurnal variations (Figure S12). The magnitudes
of transport fluxes (Figure S12) are proportional to horizontal and vertical gradients of $NO_x$ concentrations and
are therefore generally positively correlated to surface $NO_2$ concentrations. However, some exceptions exist,
reflecting different strengths of advection (U, V, and W) and turbulent mixing ($k_{zz}$) at different times. For
example, in the early morning, $NO_2$ surface concentrations peak at 5:00 – 6:00 LT (Figure 3a), while transport
fluxes peak at 7:00 – 8:00 LT (Figure S12). The delay of the peak is mainly due to lower turbulent mixing at 5:00
– 6:00 LT than other daytime hours in the model (Figure 4). Chemistry also contributes to surface $NO_2$ diurnal





variations mainly through photochemical sinks in the daytime and $N_2O_5$ hydrolysis at nighttime. Chemistry fluxes
in Figure S12 are not only correlated to the strength of photochemical reactions and $N_2O_5$ hydrolysis (chemistry
fluxes per unit $NO_x$) but are also proportional to $NO_x$ surface concentrations. Therefore, chemistry fluxes in
Figure S12 cannot directly reflect the impact of solar radiation on photochemical reactions. It can, however, still
be identified by comparing afternoon chemistry contributions: from 13:00 to 15:00 LT, surface-layer $NO_x$
emissions and $NO_2$ concentrations are increasing (Figures S12 and 3a); however, chemistry losses are decreasing
as a result of the reduction of photochemical sinks with weakening solar radiation. The contributions of vertical
mixing and photochemical sinks to $NO_2$ concentrations can be further corroborated by daytime variations of $NO_2$
vertical profiles (Figure 6) and TVCDs (Figure S13) discussed in sections 3.2 and 3.3.
Figure 3c shows the diurnal variation on weekends is also simulated well in the improved 36-km model. The
diurnal variation of surface $NO_2$ concentrations (REAM: 1.5 – 10.4 ppb; observations: 2.1 – 9.8 ppb) is lower
than on weekdays (REAM: 2.5 – 12.5 ppb; observations: 3.3 – 14.5 ppb), reflecting lower magnitude and
variation of $NO_x$ emissions on weekends (Figure 1). Figure 3d also shows an improved simulation of surface $O_3$
concentrations at nighttime due to the improved PBLH simulation (Figure 4).
*3.1.2 4-km model simulation in comparison to the surface observations*
The results of 4-km REAM simulations with original WRF $k_{zz}$ (not shown) are very similar to Figure 3 since
WRF simulated nocturnal vertical mixing is insensitive to the model horizontal resolution. Applying the modified
nocturnal mixing in the previous section also greatly reduced the nighttime $NO_2$ overestimate and $O_3$
underestimate in the 4-km REAM simulations. All the following analyses are based on REAM simulations with
improved nocturnal mixing. Figure 5 shows that mean surface $NO_2$ concentrations simulated in the 4-km model
are higher than the 36-km results over Padonia, Oldtown, Essex, Edgewood, Beltsville, and Aldino (Table S1),





leading to higher biases compared to the observations at night. A major cause is that the observation sites are
located in regions of high $NO_x$ emissions (Figures S1 and S2). At a higher resolution of 4 km, the high emissions
around the surface sites are apparent compared to rural regions. At the coarser 36-km resolution, spatial
averaging greatly reduces the emissions around the surface sites. On average, $NO_x$ emissions (molecules km$^{-2}$ s$^{-1}$)
around the six surface $NO_2$ observations sites are 67% higher in the 4-km than the 36-km REAM simulations
(Table S1). The resolution dependence of model results will be further discussed in the model evaluations using
the other in situ and remote sensing measurements.
3.2 Diurnal variations of $NO_2$ vertical profiles

Figures 6a and 6c show the temporal variations of P-3B observed and 36-km REAM simulated $NO_2$ vertical

profiles in the daytime on weekdays during the DISCOVER-AQ campaign. 36-km REAM reproduces well the
observed characteristics of $NO_2$ vertical profiles in the daytime ($R^2 = 0.91$), which are strongly affected by
vertical mixing and photochemistry (Zhang et al., 2016). When vertical mixing is weak in the early morning
(6:00 – 8:00 LT), $NO_2$, released mainly from surface $NO_x$ sources, is concentrated in the surface layer, and the
vertical gradient is large. As vertical mixing becomes stronger after 8:00 LT, $NO_2$ concentrations below 500 m
decrease significantly, while those over 500 m increase from 6:00 – 8:00 LT to 12:00 – 14:00 LT. It is
noteworthy that PBLHs and $NO_x$ emissions are comparable between 12:00 – 14:00 LT and 15:00 – 17:00 LT
(Figures 1 and 4); however, $NO_2$ concentrations at 15:00 – 17:00 LT are significantly higher than at 12:00 –
14:00 LT in the whole boundary layer, reflecting the impact of the decreased photochemical loss of $NO_x$ in the
late afternoon. In fact, photochemical losses affect all the daytime $NO_2$ vertical profiles, which can be easily
identified by $NO_2$ TVCD process diagnostics discussed in section 3.3 (Figure S13).

Figures 6b and 6d also show the observed and 36-km REAM simulated vertical profiles on weekends.

Similar to Figures 3 and 5, observed and simulated concentrations of $NO_2$ are lower on weekends than on





weekdays. Some of the variations from weekend profiles are due to a lower number of observations (47 spirals)
on weekends. The overall agreement between the observed vertical profiles and 36-km model results is good on
weekends ($R^2 = 0.81$). At 15:00 – 17:00 LT, the model simulates a larger gradient than what the combination of
aircraft and surface measurements indicates. It may be related to the somewhat underestimated PBLHs in the late
afternoon in the model (Figure 4).
On weekdays, most simulated vertical profiles at the 4-km resolution (Figure 6e) are similar to 36-km results
in part because the average $NO_x$ emissions over the six P-3B spiral sites are about the same, 4% lower in the 4-
km than the 36-km REAM simulations (Table S1). A clear exception is the 4-km REAM simulated vertical
profile at 15:00 – 17:00 LT when the model greatly overestimates boundary layer $NO_x$ mixing and
concentrations. The main reason is that WRF simulated vertical velocities ($w$) in the late afternoon are much
larger in the 4-km simulation than the 36-km simulation (Figure S11), which can explain the simulated fully
mixed boundary layer at 15:00 – 17:00 LT. Since it is not designed to run at the 4-km resolution and it is
commonly assumed that convection can be resolved explicitly at high resolutions, the convection scheme is not
used in the nested 4-km WRF simulation (Table S2); it may be related to the large vertical velocities in the late
afternoon when thermal instability is the strongest. Appropriate convection parameterization is likely still
necessary for 4-km simulations (Zheng et al., 2016), which may also help alleviate the underestimation of
precipitation in the nested 4-km WRF simulation as discussed in section 2.1.
The same rapid boundary-layer mixing due to vertical transport is present in the 4-km REAM simulated
weekend vertical profile (Figure 6f), although the mixing height is lower. Fewer spirals (47) and distinct transport
effect due to different $NO_2$ horizontal gradients between the 4-km and 36-km REAM simulations (discussed in
detail in Section 3.5) may cause the overestimation of weekend profiles in the 4-km REAM simulation.



3.3 Daytime variation of NO$_2$ TVCDs
We compare satellite, P-3B aircraft, and model-simulated TVCDs with Pandora measurements, which
provide continuous daytime observations. The locations of Pandora sites are shown in Table S1 and Figure S1.
Among the Pandora sites, four sites are located significantly above the ground level: UMCP (~20 m), UMBC
(~30 m), SERC (~40 m), and GSFC (~30 m). The other sites are 1.5 m AGL. To properly compare Pandora to
other measurements and model simulations, we calculate the missing TVCDs between the Pandora site heights
and ground surface by multiplying the Pandora TVCDs with model-simulated TVCD fractions of the
corresponding columns. The resulting correction is 2-20% ($\frac{1}{1 - missing\ TVCD\ percentage}$) for the four sites
significantly above the ground surface, but the effect on the averaged daytime TVCD variation of all Pandora
sites is small (Figure S14). In the following analysis, we use the updated Pandora TVCD data.
The weekday diurnal variations of NO$_2$ TVCDs from satellites, Pandora, 4- and 36-km REAM, and the P-3B
aircraft are shown in Figure 7a. We calculate aircraft derived TVCDs by using equation (1):
$$TVCD_{aircraft}\left(t\right) = \frac{\sum c_{aircraft}\left(t\right) \times \rho_{REAM}\left(t\right) \times V_{REAM}\left(t\right)}{A_{REAM}} \qquad (1),$$
where $t$ is time; $c_{aircraft}$ ($v/v$) denotes aircraft NO$_2$ concentrations (mixing ratios) at each level at time $t$; $\rho_{REAM}$
($molecules\ /\ cm^3$) is the density of air from 36-km REAM at the corresponding level; $V_{REAM}$ ($cm^3$) is the volume of
the corresponding 36-km REAM grid cell; $A_{REAM}$ ($cm^2$) is the surface area ($36 \times 36$ km$^2$). In the calculation, we
only use NO$_2$ concentrations below 3.63 km AGL because few aircraft measurements were available above this
height in the campaign. Missing tropospheric NO$_2$ above 3.63 km AGL in the aircraft TVCD calculation has little
impact on our analyses, as 36-km REAM model simulation shows that 85% ± 12% of tropospheric NO$_2$ are
located below 3.63 km AGL during 6:00 – 17:00 LT in the DISCOVER-AQ region, which is roughly consistent
with the GMI model results with 85% - 90% tropospheric NO$_2$ concentrated below 5 km (Lamsal et al., 2014). It



should be noted that only six P-3B spirals are available during the campaign, less than the samplings of 11 inland
Pandora sites.

The 4-km REAM simulated NO₂ TVCDs are mostly higher than the 36-km results and the observations

(Figure 7a). However, since the standard deviations of the data are much larger than the model difference, the 4-
and 36-km model results show generally similar characteristics relative to the observations. REAM simulation
results are in reasonable agreement with Pandora, P-3B aircraft, and satellite daytime NO₂ TVCDs, except that
NASA-derived OMI (OMNO2) TVCDs are somewhat lower than other datasets, which may be partly due to
biased a priori vertical profiles from the GMI model in the NASA retrieval in the campaign (Lamsal et al., 2014;
Lamsal et al., 2020). TVCDs derived by using the DOMINO algorithm and 36-km REAM NO₂ vertical profiles
are in agreement with those from KNMI, which indicates that the TM4 model from KNMI provides reasonable
estimates of a priori NO₂ vertical profiles on weekdays in the DISCOVER-AQ region in summer.

We find evident decreases of NO₂ TVCDs from GOME-2A to OMI in Figure 7a, which is consistent with

Pandora, REAM results, and previous studies that showed decreasing NO₂ TVCDs from SCIAMACHY to OMI
due to photochemical losses in summer (Boersma et al., 2008; Boersma et al., 2009). P-3B aircraft TVCDs also
show this decrease feature but have large variations due in part to the limited aircraft sampling data.

Pandora NO₂ TVCD data have different characteristics from REAM simulated and P-3B aircraft measured

TVCDs at 5:00 – 7:00 LT and 14:00 – 18:00 LT (Figure 7a). At 5:00 – 7:00 LT, Pandora data show a significant
increase of NO₂ TVCDs, but REAM and aircraft TVCDs decrease. At 14:00 LT – 18:00 LT, Pandora TVCDs
have little variations, but REAM and aircraft TVCDs increase significantly. The relatively flat Pandora TVCDs
in the late afternoon compared to REAM and P-3B aircraft measurements are consistent with Lamsal et al.
(2017), which found the significant underestimation (26% – 30%) of Pandora VCDs compared to UC-12 ACAM





measurements from 16:00 LT to 18:00 LT during the DISCOVER-AQ campaign. We show the simulated effects
of emission, chemistry, transport, and dry deposition on $NO_x$ TVCDs in Figure S13. The simulated early morning
decrease of $NO_2$ TVCDs is mainly due to the chemical transformation between $NO_2$ and NO favoring the
accumulation of NO under low-$O_3$ and low-$HO_2/RO_2$ conditions, thus NO TVCDs increase significantly but $NO_2$
TVCDs continue decreasing during the period. The increase in the late afternoon is primarily due to the decrease
of photochemistry-related sinks. The reasons for the discrepancies of $NO_2$ TVCDs between Pandora and REAM
results during the above two periods are unclear. Large SZAs in the early morning and the late afternoon (Figure
S8) lead to the higher uncertainties of Pandora measurements (Herman et al., 2009), although we have excluded
Pandora measurements with SZA > 80°. In addition, Pandora is a sun-tracking instrument with a small effective
FOV and is sensitive to local conditions within a narrow spatial range which may differ significantly from the
average properties of 36- and 4-km grid cells depending upon the time of the day (Figure S9) (Herman et al.,
2009; Herman et al., 2018; Herman et al., 2019; Judd et al., 2018; Judd et al., 2019; Judd et al., 2020; Lamsal et
al., 2017; Reed et al., 2015). Another possible reason is that Pandora instruments had few observations in the
early morning, and the resulting average may not be representative (Figure S9).
To further understand the daytime variation of $NO_2$ TVCDs, we examine P-3B aircraft data derived and
REAM simulated $NO_2$ VCD variations for different height bins (Figure 8). $NO_2$ VCDs below 3.63 km AGL
display a "U"-shaped pattern from 5:00 LT to 17:00 LT. In the morning, as vertical mixing becomes stronger
after sunrise, high-$NO_x$ air in the lower layer is mixed with low-$NO_x$ air in the upper layer. The increase of $NO_x$
vertical mixing above 400 m is sufficient to counter the increase of photochemical loss in the morning.
Conversely, the $NO_2$ VCDs below 400 m decrease remarkably from sunrise (about 6:00 LT) to around noontime
due to both vertical mixing and the increase of photochemical strength. From 13:00 LT to 16:00 LT, $NO_2$ VCDs
increase slowly, reflecting a relative balance among emissions, transport, chemistry, and dry depositions. The
sharp jump of the VCDs from 16:00 LT to 17:00 LT is mainly due to dramatically reduced chemical loss. And 4-





km REAM simulated $NO_2$ VCDs at 0.40-3.63 km at 16:00-17:00 LT are much higher than 36-km results because
of the rapid vertical mixing in the 4-km REAM simulation (Figures 6 and S11).
Similar to $NO_2$ surface concentrations and vertical profiles in Figures 5 and 6, the $NO_2$ TVCD variation is
also smaller on weekends than on weekdays, but the day-night pattern is similar (Figure 7). Although the 4-km
REAM $NO_2$ TVCDs are generally higher than the 36-km results and observations in the daytime, considering
their large standard deviations, $NO_2$ TVCDs from both simulations are comparable to satellite products, Pandora,
and P-3B aircraft observations most of the time on weekends. The exception is that Pandora TVCDs have much
less variation in the early morning and late afternoon than REAM simulation and aircraft datasets. Another
anomaly is that KNMI GOME-2A TVCDs at 9:30 LT are much larger than the other datasets, while the GOME-
2A TVCDs retrieved using 36-km REAM profiles shows comparable TVCDs to Pandora, REAM, and aircraft
datasets, reflecting possible biased $NO_2$ a priori profiles from the TM4 model on weekends used in the KNMI
GOME-2A retrieval.
3.4 Model comparisons with $NO_y$ measurements
$NO_y$ is longer-lived than $NO_x$, and $NO_y$ concentrations are not affected by chemistry as much as $NO_x$. We
obtain two types of $NO_y$ concentrations from the P-3B aircraft in the DISCOVER-AQ campaign: one is $NO_y$
concentrations directly measured by the NCAR 4-channel instrument, corresponding to the sum of NO, $NO_2$,
$\sum$PNs, $\sum$ANs, $HNO_3$, $N_2O_5$, $HNO_4$, HONO, and the other reactive nitrogenic species in REAM (all the other
species are described in Table 1); the other one, which we name as "derived-$NO_y$", is the sum of NO from the
NCAR 4-channel instrument and $NO_2$ ($NO_2$_LIF), $\sum$PNs, $\sum$ANs, and $HNO_3$ measured by the TD-LIF technique,
corresponding to NO, $NO_2$, $\sum$PNs, $\sum$ANs, and $HNO_3$ in REAM (Table 1). On average, P-3B derived-$NO_y$
concentrations (2.88 ± 2.24 ppb) are 17% higher than coincident P-3B $NO_y$ concentrations (2.46 ± 2.06 ppb) with
$R^2 = 0.75$, generally reflecting consistency between these two types of measurements. As shown in Table 1, on





weekdays, the 36-km REAM $NO_y$ concentrations are 45% larger than P-3B with $R^2 = 0.33$, and the 36-km
REAM derived-$NO_y$ concentrations are 8% larger than P-3B with $R^2 = 0.41$. 4-km REAM show similar results,
suggesting that REAM simulations generally reproduce the observed $NO_y$ and derived-$NO_y$ concentrations within
the uncertainties, although the average values from REAM are somewhat larger than the observations. The
concentrations of weekday NO, $NO_2$, and $\sum$PNs from REAM simulations are also comparable to the
observations. However, weekday $\sum$ANs concentrations are 68% lower in the 36-km REAM than observations,
suggesting that the chemistry mechanism in REAM may need further improvement to better represent isoprene
nitrates. It is noteworthy that, since $\sum$ANs only account for a small fraction (~11%) in observed derived-$NO_y$, the
absolute difference between REAM simulated and P-3B observed $\sum$ANs concentrations is still small compared to
$HNO_3$. Weekday $HNO_3$ concentrations are significantly higher in REAM simulations (36-km: 57%, 0.65 ppb; 4-
km: 70%, 0.82 ppb) than P-3B observations, which is the main reason for the somewhat larger $NO_y$ and derived-
$NO_y$ concentrations in REAM compared to P-3B observations. The higher $HNO_3$ concentrations in REAM may
be related to the underestimation of precipitation in the corresponding WRF simulations, as discussed in section
2.1 (Figures S5), leading to the underestimated wet scavenging of $HNO_3$, especially for the 4-km REAM
simulation.
We also examine the weekday diurnal variations of derived-$NO_y$ vertical profiles from P-3B and REAM
simulations in Figure S15. Generally, both 36- and 4-km REAM simulations capture the variation characteristics
of observed vertical profiles, which are similar to those for $NO_2$ in Figure 6. REAM derived-$NO_y$ concentrations
are comparable to P-3B observations at most vertical levels on weekdays. Some larger derived-$NO_y$
concentrations in the model results can be partially explained by larger $HNO_3$ concentrations in REAM, such as
those below 1 km at 9:00 – 11:00 LT for the 36-km REAM and those below 2.0 km at 12:00 – 17:00 LT for the
4-km REAM (Figure S16).





Figure 9 shows the comparison of the diurnal cycles of surface $NO_y$ concentrations observed at Padonia,
Edgewood, Beltsville, and Aldino during the DISCOVER-AQ campaign with those from the REAM simulations.
Generally, the REAM simulations reproduce the observed surface $NO_y$ diurnal cycles except for the spikes
around 17:00 – 20:00 LT due to still underestimated PBLHs (Figure 4). 4-km simulation results have a higher
bias than 36-km results relative to the observations, similar to the comparisons of $NO_2$ surface concentrations and
TVCDs in Figures 5 and 7 due to higher emissions around the observation sites in 4- than 36-km simulations
(Table S1 and Figure S2).
3.5 Resolution dependence of $NO_x$ emission distribution
We show previously that the 4-km REAM simulated $NO_2$ and $NO_y$ surface concentrations and $NO_2$ TVCDs
are higher than observations at daytime in comparison to the corresponding 36-km REAM results. An
examination of monthly mean $NO_2$ surface concentrations and TVCDs for July 2011 also shows that 4-km
simulation results are significantly higher than the 36-km results over the 11 inland Pandora sites in the daytime
(Figure S17). The process-level diagnostics in Figure S13 indicate that the mean contribution of $NO_x$ emissions
to $NO_x$ ΔTVCDs in the 4-km simulation is $1.32 \times 10^{15}$ molecules $cm^{-2}$ $h^{-1}$ larger than that in the 36-km simulation
between 9:00 LT and 16:00 LT, while the absolute mean contributions of chemistry and transport (they are
negative in Figure S13, so we use absolute values here) in the 4-km simulation are $0.22 \times 10^{15}$ and $0.99 \times 10^{15}$
molecules $cm^{-2}$ $h^{-1}$ larger than the 36-km simulation, respectively. The contributions of dry deposition to $NO_x$
ΔTVCDs are negligible compared to other factors in both simulations (Figure S13). Therefore, the 34% higher
$NO_x$ emissions over the 11 inland Pandora sites (Table S1 and Figure 1) is the main reason for the larger daytime
$NO_2$ surface concentrations and TVCDs in the 4-km than the 36-km REAM simulations (Figure S17). The
significantly different contribution changes between $NO_x$ emissions ($1.32 \times 10^{15}$ molecules $cm^{-2}$ $h^{-1}$ or about one
third) and chemistry ($0.22 \times 10^{15}$ molecules $cm^{-2}$ $h^{-1}$ or about 7%) reflect potential chemical nonlinearity (Li et al.,
2019; Silvern et al., 2019; Valin et al., 2011) and transport effect. Different transport contributions between the 4-





km and the 36-km REAM are mainly caused by their different $NO_x$ horizontal gradients (Figures S2 and 10),
while the impact of wind fields is small since we do not find significant differences in horizontal wind
components between the two simulations (Figure S18). The impact of transport on the two simulations can be
further verified by the comparison of $NO_2$ TVCDs over the six P-3B spiral sites between the two simulations
(Figure S19). Mean $NO_x$ emissions over the six P-3B spiral sites are close (relative difference $< 4\%$) between the
two simulations (Table S1 and Figure S19). From 9:00 to 12:00 LT, the contributions of $NO_x$ emissions to $NO_x$
ΔTVCDs are $2.50 \times 10^{15}$ and $2.49 \times 10^{15}$ molecules $cm^{-2}$ $h^{-1}$ for the 36-km and 4-km REAM simulations,
respectively, and the contributions of chemistry are also close between the two simulations (36-km: $-2.64 \times 10^{15}$
molecules $cm^{-2}$ $h^{-1}$; 4-km: $-2.68 \times 10^{15}$ molecules $cm^{-2}$ $h^{-1}$). However, the contributions of transport are $-0.32 \times$
$10^{15}$ and $0.04 \times 10^{15}$ molecules $cm^{-2}$ $h^{-1}$ for the 36-km and 4-km REAM simulations, respectively, leading to
larger $NO_2$ TVCDs in the 4-km REAM simulation than the 36-km REAM from 9:00 – 12:00 LT (Figure S19c).
Since horizontal wind fields over the six P-3B spiral sites are comparable between two simulations (Figures S3,
S4, and S18) and larger $NO_x$ horizontal gradients are found near the P-3B spiral sites for the 4-km REAM (Figure
S2), we attribute the different transport contributions between the two simulations to a much larger $NO_x$ emission
gradient around the measurement locations in 4-km than 36-km emission distributions.
We re-grid the 4-km REAM results into the grid cells of the 36-km REAM, which can significantly reduce
the impact of different $NO_x$ emission distributions and associated transport on the two simulations. Compared to
the original 4-km REAM results, the re-gridded surface $NO_2$ concentrations and TVCDs over the 11 inland
Pandora sites are much closer to the 36-km REAM results (Figure S17). After re-gridding the 4-km REAM
results into 36-km REAM grid cells, we also find more comparable $NO_y$ surface concentrations between the re-
gridded 4-km results and the 36-km REAM results (Figure S20). The remaining discrepancies between the re-
gridded results and the 36-km REAM results may be due to chemical nonlinearity and other meteorological
effects, such as larger vertical wind in the 4-km REAM (Figure S11) and their different $k_{zz}$ values in the PBL.



Although other factors, such as chemical nonlinearity and vertical diffusion, may affect the 36-km and 4-km
REAM simulations differently, the difference between 4- and 36-km simulations of reactive nitrogen is largely
due to that of $NO_x$ emissions.

The 4- and 36-km simulation difference depends on the location of the observations. In some regions, the

$NO_x$ emission difference between 4- and 36-km simulations is small. The comparison of $NO_y$ measurements from
P-3B spirals with coincident REAM results in Table 1 suggests that the 4-km and 36-km REAM simulations
produce similar $NO_y$ (relative difference ~2%) and derived-$NO_y$ (relative difference ~4%) concentrations on
weekdays, and both simulation results are comparable to the observations. The $NO_y$ similarity over the P-3B
spiral sites between the 36-km and 4-km REAM simulations is consistent with the comparable $NO_x$ emissions
over (relative difference $< 4\%$) the six P-3B spiral sites between the two simulations (Table S1). The differences
between the 4-km model simulation results and P3-B observations are larger on weekends than on weekdays
(Table 1) due to the limited weekend sampling since model simulated monthly mean values show similar
differences between the 4-km and 36-km REAM simulations on weekends as on weekdays (not shown).
3.6 Evaluation of 4-km $NO_x$ distribution with ACAM measurements

The evaluation of model simulations of surface, aircraft, and satellite observations tends to point out a high

bias in 4- than 36-km model simulations. However, we note that the uncertainties of the observations and model
data are often comparable or larger than the model differences. Here we examine the 4-km model simulated $NO_2$
VCDs with high-resolution ACAM measurements onboard the UC-12 aircraft in Figures 10 and S21,
respectively. The spatial distributions of ACAM and 4-km REAM $NO_2$ VCDs are generally consistent with $R^2 =$
0.37 on weekdays and $R^2 = 0.50$ on weekends. The domain averages of ACAM and 4-km REAM $NO_2$ VCDs are
$4.7 \pm 2.0$ and $4.5 \pm 3.2 \times 10^{15}$ molecules cm$^{-2}$ on weekdays and $3.0 \pm 1.7$ and $3.3 \pm 2.8 \times 10^{15}$ molecules cm$^{-2}$ on
weekends, respectively. The spatial distributions of ACAM and 4-km REAM $NO_2$ VCDs are highly correlated



with the spatial distribution of 4-km NEI2011 $NO_x$ emissions. All three distributions capture two strong peaks
around Baltimore and Washington, D.C. urban regions and another weak peak in the northeast corner of the
domain (Wilmington city in Delaware) (Figures 10 and S21). However, Figures 10 and S21 clearly show that
$NO_2$ VCDs from the 4-km REAM simulation are more concentrated in Baltimore and Washington, D.C. urban
regions than ACAM, which are also reflected by the higher $NO_2$ VCD standard deviations of the 4-km REAM
results than ACAM. Several Pandora sites are in the highest $NO_2$ VCD regions where the 4-km REAM generally
produces larger $NO_2$ VCDs than ACAM, which explains why the $NO_2$ TVCDs over the 11 Pandora sites from the
4-km REAM simulation are higher than the observations (Figure 7) and the 36-km REAM results (Figure S17)
around noontime. Horizontal transport cannot explain the $NO_2$ VCD distribution biases in the 4-km REAM
simulation due to the following reasons. Firstly, horizontal wind fields are simulated as well by the nested 4-km
WRF simulation as the 36-km WRF compared to P-3B measurements, as discussed in section 2.1. Secondly, the
prevailing northwest wind in the daytime (Figure S4) should move $NO_x$ eastward, but we find no significant
eastward shift of $NO_2$ VCDs compared to $NO_x$ emissions in both ACAM and 4-km REAM distributions (Figure
10). Lastly, we find a local minimum of $NO_2$ VCDs in the middle of the Baltimore urban region (the purple circle
in Figure 10b) in the ACAM distribution, which cannot be explained by horizontal transport or chemical
nonlinearity due to the surrounding high $NO_x$ emissions in the 4-km REAM simulation. Therefore, we attribute
the distribution inconsistency between ACAM and the 4-km REAM to the distribution biases of NEI2011 $NO_x$
emissions at the 4-km resolution since the average below-aircraft $NO_2$ VCDs between ACAM and the 4-km
REAM are about the same.

It is noteworthy that about 91% ACAM $NO_2$ VCD data are measured from 8:00 – 16:00 LT, and only using

ACAM $NO_2$ VCDs between 8:00 and 16:00 LT for the above comparison does not affect our results shown here.
Moreover, to minimize the effect of overestimated afternoon vertical mixing (Figure 6) on the 4-km REAM





simulation results, we also examine the comparison between ACAM NO$_2$ VCDs from 9:00 – 14:00 LT with
coincident 4-km REAM results, which produces similar results as shown here.

We also evaluate the NO$_2$ VCD distributions from the 4-km REAM simulation on weekdays and weekends

with ACAM NO$_2$ VCDs below the U-12 aircraft obtained from https://www-air.larc.nasa.gov/cgi-
bin/ArcView/discover-aq.dc-2011?UC12=1#LIU.XIONG/ in Figures S22 and S23. Although the domain mean
ACAM NO$_2$ VCDs in Figures S22 and S23 are higher than coincident 4-km REAM results due to the different
retrieval method from Lamsal et al. (2017), such as different above-aircraft NO$_2$ VCDs and different a priori NO$_2$
vertical profiles, we can still find clear distribution inconsistencies between the 4-km REAM and ACAM NO$_2$
VCDs. The 4-km REAM NO$_2$ VCDs are more concentrated in the Baltimore and Washington, D.C. urban regions
than this set of ACAM data, which is consistent with the conclusions derived from the ACAM dataset retrieved
by Lamsal et al. (2017).
3.7 Implications for NO$_x$ emissions

The analysis of section 3.6 indicates that the NEI2011 NO$_x$ emission distribution at the 4-km resolution is

likely biased for the Baltimore-Washington region. The distribution bias of the high-resolution NO$_x$ emission
inventories is corroborated by the comparison of the NO$_x$ emission inventory derived from the CONsolidated
Community Emissions Processor Tool, Motor Vehicle (CONCEPT MV) v2.1 with that estimated by the Sparse
Matrix Operator Kernel Emissions (SMOKE) v3.0 model with the Motor Vehicle Emissions Simulator (MOVES)
v2010a (DenBleyker et al., 2012). CONCEPT with finer vehicle activity information as input produced a wider-
spread but less-concentrated running exhaust NO$_x$ emissions compared to MOVES in the Denver urban area for
July 2008 (DenBleyker et al., 2012). In addition, Canty et al. (2015) found that CMAQ 4.7.1, with on-road
emissions from MOVES and off-road emissions from the National Mobile Inventory Model (NMIM),
overestimated NO$_2$ TVCD over urban regions and underestimated NO$_2$ TVCDs over rural areas in the





northeastern U.S. for July and August 2011 compared to the OMNO2 product. The urban-rural contrast was also
found in Texas during the 2013 DISCOVER-AQ campaign in the studies of Souri et al. (2016) and Souri et al.
(2018), implying distribution uncertainties in $NO_x$ emissions, although these studies and Canty et al. (2015)
focused more on polluted regions with overestimated $NO_x$ emissions in their conclusions. The emission
distribution bias may also explain why Anderson et al. (2014) have different results from our simulated
concentrations in Table 1. In their study, they compared in-situ observations with a nested CMAQ simulation
with a resolution of 1.33 km. It is difficult to build up a reliable emission inventory for the whole U.S. at very
high resolutions with currently available datasets due to the significant inhomogeneity of $NO_x$ emissions Marr et
al. (2013), but we can still expect significant improvements of the temporal-spatial distributions of $NO_x$
emissions in the near future as GPS-based information start to be used in the NEI estimates (DenBleyker et al.,

2017).

Although the NEI2011 $NO_x$ emission distribution is likely biased at 4-km resolution, the similar average
below-aircraft $NO_2$ VCDs between ACAM and the 4-km REAM (Figure 10), as well as the good performance of
the 36-km REAM compared to $NO_2$ and $NO_y$ observations (Figures 3 and 5-9 and Table 1), suggest that NEI2011
may provide reliable estimates of $NO_x$ emissions over the Baltimore-Washington region at coarser resolutions. It
should be noted that although the good performance of the 36-km REAM in reproducing P-3B, Pandora, and
surface observations may be limited by representative errors due to the relatively coarse resolution of 36-km
REAM, ACAM and satellite $NO_2$ VCDs are all observations covering large areas, alleviating the problem caused
by observations limited to small areas. Our conclusion on the potential reliability of NEI2011 is consistent with
Salmon et al. (2018), who found NEI2011 and NEI2014 were in agreement with aircraft observation-derived $NO_x$
emissions during the Wintertime INvestigation of Transport, Emissions, and Reactivity (WINTER) campaign in
February – March 2015 around the Washington, D.C.-Baltimore area. The agreement was further confirmed
through the investigation of observed and NEI $NO_x/CO_2$, $CO/NO_x$, and $CO/CO_2$ ratios (Salmon et al., 2018).





However, our evaluation of NEI $NO_x$ emissions is different from Travis et al. (2016) and Anderson et al. (2014).
Travis et al. (2016) compared the GEOS-Chem simulation results with the observations of $NO_x$ and its oxidation
products from the SEAC[4]RS campaign in the Southeast US, nitrate wet deposition fluxes from the National Acid
Deposition Program (NADP) network, and $NO_2$ TVCDs from OMI, and suggested that NEI2011 overestimated
mobile and industrial $NO_x$ emissions by $30\% - 60\%$. The GEOS-Chem chemical mechanism from Travis et al.
(2016) is almost the same as what we use in REAM, and their model simulation has a horizontal resolution of
$0.25° \times 0.3125°$, which is also close to REAM (36 km $\times$ 36 km). We attribute the discrepancies between Travis et
al. (2016) and our study to the regional discrepancies. Travis et al. (2016) derived their conclusions based on the
averages of large domains (the whole CONUS or the Southeast U.S.), while our study focuses on the much
smaller Baltimore-Washington region. If limited to the Baltimore-Washington region, Figures 3 and 5 in Travis
et al. (2016) shown that nitrate wet deposition fluxes from NADP and TVCDs from the Berkeley High-
Resolution (BEHR) retrieval and NASA (OMNO2) were significantly higher than their simulations with NEI
non-power-plant $NO_x$ emissions reduced by $60\%$, implying their conclusions about the overestimation of NEI
$NO_x$ emissions at least not applicable to the Baltimore-Washington region. Anderson et al. (2014) evaluated
NEI2011 emissions with the observed concentration ratios of CO to $NO_y$ and CO to $NO_x$ from the same
DISCOVER-AQ campaign. They concluded that NEI overestimated $NO_x$ emissions by 51% - 70% in Maryland
in the summer of 2011. However, the uncertainties of scaling the emission ratios of CO/$NO_x$ to the concentrations
ratios of CO/$NO_y$ or CO/$NO_x$ can be large due in part to the large contribution of biogenic isoprene oxidation to
the variation of CO (Cheng et al., 2017). Furthermore, base CMAQ-simulated $\sum$PNs (1.4 ppbv) and $\sum$ANs (0.96
ppbv) concentrations by Anderson et al. (2014) are 0.79 and 0.64 ppbv higher than the corresponding P-3B
observations, respectively, leading to the overestimation of $NO_y$ in their model; these biases are much larger than
those (~0.2 ppbv) from our 36-km REAM simulation as shown in Table 1, possibly due to different chemical
mechanisms used by CMAQ and REAM. Another possible reason, as mentioned above, is that the base CMAQ


used by Anderson et al. (2014) has a horizontal resolution of 1.33 km, much higher than our REAM simulations;
$NO_x$ emission distribution may be a potential issue causing the overestimation of the base CMAQ results.
Here, we emphasize that our study is not necessarily contradictory to recent studies concerning the
overestimation of NEI $NO_x$ emissions (Anderson et al., 2014; Canty et al., 2015; McDonald et al., 2018; Souri et
al., 2016; Souri et al., 2018; Travis et al., 2016). Different types of observations in different periods and locations
are analyzed for various purposes. This study focuses more on the spatial distribution of $NO_x$ emissions in
NEI2011 at the 4-km resolution, while previous studies are concerned more about the $NO_x$ emission magnitudes
in highly polluted sites, although the spatial distribution issue was also mentioned in some of the studies. If we
limit our analyses to those observations in Figures 5, 7, and 9 and the 4-km REAM, we would also conclude an
overestimation of NEI $NO_x$ emissions. In this study, through comprehensive evaluations of $NO_2$ and $NO_y$
measurements at 36- and 4-km resolutions, we provide another possible explanation for the overestimation of
high-resolution model results at polluted sites. Although our assessment of NEI2011 $NO_x$ emissions at 36-km
resolution is contingent upon potential representative errors, the further comparisons of below-aircraft $NO_2$ VCDs
between the 4-km REAM and ACAM from Lamsal et al. (2017) suggest that total $NO_x$ emissions seem accurate
but the emission distribution seems biased in the 4-km NEI2011 in our defined domain (Figure 10). Considering
the significant heterogeneity of $NO_x$ emissions, different conclusions on $NO_x$ emission biases may be made in
other regions, but the spatial distribution of $NO_x$ emissions is a critical factor in evaluating $NO_x$ emissions and
improving emission estimation models, which deserves more attention in future studies.

## 4 Conclusions


We investigate the diurnal cycles of surface $NO_2$ concentrations, $NO_2$ vertical profiles, and $NO_2$ TVCDs
using REAM model simulations on the basis of the observations from air quality monitoring sites, aircraft,
Pandora, OMI, and GOME-2A during the DISCOVER-AQ 2011 campaign. We find that WRF simulated





nighttime PBLHs are significantly lower than ELF lidar measurements. Increasing nighttime mixing from 18:00
– 5:00 LT in the REAM simulations, we significantly improve REAM simulations of nighttime surface $NO_2$ and
$O_3$ concentrations.

The REAM simulation reproduces well the observed diurnal cycles of surface $NO_2$ and $NO_y$ concentrations,

$NO_2$ vertical profiles, and $NO_2$ TVCDs on weekdays. Observed $NO_2$ concentrations in the boundary layer and
TVCDs on weekends are significantly lower than on weekdays. By specifying a weekend to weekday $NO_x$
emission ratio of 2:3 and applying a less variable $NO_x$ emission diurnal profile on weekends than weekdays,
REAM can simulate well the weekend observations. A few issues are also noted. First, Pandora TVCDs show
different variations from aircraft-derived and REAM-simulated TVCDs in the early morning and late afternoon,
which may be due to the uncertainties of Pandora measurements at large SZAs and the small effective FOV of
Pandora. Second, the weekday OMI $NO_2$ TVCDs derived by NASA are somewhat lower than the KNMI OMI
product, P-3B aircraft-derived TVCDs, Pandora, and REAM results; the difference may be caused by the a priori
vertical profiles used in the NASA retrieval. Lastly, the weekend OMI $NO_2$ TVCDs derived by KNMI are larger
than those from Pandora, P-3B aircraft, REAM, and the OMI retrieval with REAM $NO_2$ vertical profiles,
indicating a possible bias of the TM4 simulated a priori $NO_2$ vertical profiles in the weekend mornings during
DISCOVER-AQ 2011.

While a higher-resolution simulation is assumed to be superior at a priori, the large observation dataset

during DISCOVER-AQ 2011 offers the opportunity of a detailed comparison of 4-km and 36-km model
simulations. In general, the 4-km simulation results tend to have a high bias relate to the 36-km results in light of
the observations. We find two areas that have not been widely recognized for high-resolution model simulations.
The first is not using convection parameterization in high-resolution WRF simulations since convection can be
resolved explicitly and convection parameterizations are not designed for high-resolution simulations. We find



that 4-km WRF tends to overestimate boundary-layer mixing and vertical transport in the late afternoon, leading
to a high model bias in simulated $NO_2$ vertical profiles compared to P-3B aircraft observations. The reasons for
this late-afternoon bias in 4-km WRF simulations and model modifications to mitigate this bias need further
studies.

A second issue is related to the spatial distribution of $NO_x$ emissions in NEI2011. At 4-km, the grid cells

over the 11 inland Pandora sites have 34% higher $NO_x$ emissions than the 36-km grid cells. Consequently, 4-km
REAM overestimates $NO_2$ concentrations and TVCDs than the observations. The 4-km grid cells over four
surface $NO_y$ measurement sites have about a factor of 2 higher $NO_x$ emissions than the corresponding 36-km grid
cells, leading to significantly overestimated $NO_y$ concentrations in the 4-km REAM simulation compared to the
observations. After we re-grid the 4-km $NO_2$ and $NO_y$ results to the 36-km grid cells, the results of 4-km
simulations are similar to the original 36-km simulations. The comparison of 4-km ACAM $NO_2$ VCD
measurements from the UC-12 aircraft with coincident 4-km REAM results shows that 4-km REAM $NO_2$ VCDs
are more concentrated in urban regions than the ACAM observations. Further model analysis indicates that the 4-
km VCD discrepancies are due primarily to the distribution bias of 4-km NEI2011 $NO_x$ emissions. At high
resolutions, potential biases in the emission inventories are accentuated when model results are evaluated with the
observations. Our results highlight the research need to improve the methodologies and datasets used in emission
estimates at high resolutions.
**Data availability**
The DISCOVER-AQ 2011 campaign datasets are archived on https://www-air.larc.nasa.gov/cgi-
bin/ArcView/discover-aq.dc-2011 (last access: March 6, 2020). EPA air quality monitoring datasets are from
https://www3.epa.gov/airdata/ (last access: June 23, 2015). The NASA OMI $NO_2$ product is from
https://disc.gsfc.nasa.gov/datasets/OMNO2_003/summary (last access: September 26, 2020). The KNMI OMI



NO$_2$ product is from http://www.temis.nl/airpollution/no2.html (last access: January 14, 2015). We obtain the
KNMI GOME-2A NO$_2$ VCD archives from http://www.temis.nl/airpollution/no2col/no2colgome2_v2.php (last
access: January 22, 2015). The GMI MERRA-2 simulation results are from
https://portal.nccs.nasa.gov/datashare/dirac/gmidata2/users/mrdamon/Hindcast-
Family/HindcastMR2/2011/stations/ (last access: May 14, 2019). We obtain the UC-12 ACAM NO$_2$ VCD
product by X. Liu from https://www-air.larc.nasa.gov/cgi-bin/ArcView/discover-aq.dc-
2011?UC12=1#LIU.XIONG/ (last access: December 31, 2019). The Stage IV precipitation data is downloaded
from https://rda.ucar.edu/datasets/ds507.5/ (last access: December 28, 2019). The NCEP CFSv2 6-hourly product
is available at http://rda.ucar.edu/datasets/ds094.0/ (last access: March 10, 2015). REAM simulation results for
this study and the UC-12 ACAM NO$_2$ VCD product by Lamsal et al. (2017) are available upon request.
**Author contribution**
JL and YW designed the study. JL, RZ, and CS updated the REAM model. JL conducted model simulations.
KFB developed the DOMINO algorithm, CS applied the algorithm to REAM vertical profiles, and JL updated the
retrieval algorithm and did the retrieval by using REAM NO$_2$ vertical profiles. AW, JH, EAC, RWL, JJS, RD,
AMT, TNK, LNL, SJJ, MGK, XL, CRN made various measurements in the DISCOVER-AQ 2011 campaign. JL
conducted the analyses with discussions with YW, RZ, CS, AW, JH, KFB, EAC, RWL, JJS, RD, AMT, TNK,
LNL, SJJ, MGK, XL, and CRN. JL and YW led the writing of the manuscript with inputs from all other
coauthors. All coauthors reviewed the manuscript.
**Competing interests**
The authors declare that they have no conflict of interest.





**Acknowledgments**


This work was supported by the NASA ACMAP Program. We thank Chun Zhao for providing us the PNNL
NEI2011 emission inventory. We thank Yuzhong Zhang and Jenny Fisher for providing the updated GEOS-
Chem chemistry mechanism files and thank Yuzhong Zhang, Yongjia Song, Hang Qu, Ye Cheng, Aoxing Zhang,
Yufei Zou and Ziming Ke for discussion with J. Li. We thank Susan Strahan for providing the GMI outputs
download link.

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

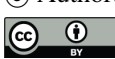

**Table 1.** Comparison of the concentrations of $NO_y$ and its components between REAM and P-3B aircraft measurements during the DISCOVER-AQ campaign

| | | | $NO_y$ / ppb[1] | NO / ppb | NO2_NCAR / ppb | NO2_LIF / ppb[2] | ∑PNs / ppb | ∑ANs / ppb | HNO3 / ppb | Derived-NOy / ppb[3] |
|---|---|---|---|---|---|---|---|---|---|---|
| 36-km[4] | Weekday[5] | P-3B | 2.51 ± 2.09 | 0.18 ± 0.29 | 0.85 ± 1.13 | 0.68 ± 0.95 | 0.70 ± 0.58 | 0.31 ± 0.23 | 1.15 ± 0.73 | 2.86 ± 2.26 |
| | | REAM | 3.64 ± 3.15 | 0.18 ± 0.31 | 0.74 ± 1.06 | 0.68 ± 0.90 | 0.53 ± 0.45 | 0.10 ± 0.09 | 1.80 ± 1.62 | 3.10 ± 2.71 |
| | | R² | 0.33 | 0.33 | 0.37 | 0.33 | 0.37 | 0.38 | 0.24 | 0.41 |
| | Weekend | P-3B | 3.01 ± 2.19 | 0.15 ± 0.20 | 0.71 ± 0.80 | 0.63 ± 0.72 | 0.91 ± 0.53 | 0.36 ± 0.21 | 1.15 ± 0.79 | 2.96 ± 2.15 |
| | | REAM | 3.76 ± 2.24 | 0.15 ± 0.17 | 0.53 ± 0.61 | 0.53 ± 0.60 | 0.52 ± 0.29 | 0.09 ± 0.06 | 2.30 ± 1.40 | 3.41 ± 2.30 |
| | | R² | 0.29 | 0.28 | 0.41 | 0.45 | 0.27 | 0.38 | 0.49 | 0.51 |
| 4-km | Weekday | P-3B | 2.51 ± 2.15 | 0.19 ± 0.30 | 0.85 ± 1.29 | 0.67 ± 0.96 | 0.70 ± 0.59 | 0.31 ± 0.22 | 1.17 ± 0.74 | 2.90 ± 2.27 |
| | | REAM | 3.67 ± 3.63 | 0.17 ± 0.29 | 0.74 ± 1.12 | 0.72 ± 1.10 | 0.45 ± 0.51 | 0.08 ± 0.10 | 1.99 ± 1.92 | 3.23 ± 3.24 |
| | | R² | 0.25 | 0.46 | 0.48 | 0.55 | 0.37 | 0.27 | 0.60 | 0.44 |
| | Weekend | P-3B | 2.97 ± 2.13 | 0.15 ± 0.18 | 0.69 ± 0.75 | 0.65 ± 0.85 | 0.90 ± 0.51 | 0.35 ± 0.21 | 1.15 ± 0.79 | 2.93 ± 2.08 |
| | | REAM | 4.22 ± 3.79 | 0.24 ± 0.42 | 0.83 ± 1.40 | 0.78 ± 1.34 | 0.38 ± 0.27 | 0.07 ± 0.08 | 2.48 ± 2.04 | 3.59 ± 3.59 |
| | | R² | 0.20 | 0.33 | 0.37 | 0.30 | 0.15 | 0.20 | 0.59 | 0.35 |

[1] For P-3B, the concentrations of $NO_y$, NO, and NO2_NCAR were measured by using the NCAR 4-channel chemiluminescence instrument. The measurement uncertainties are 10%, 10 - 15%, and 10% for NO, NO2, and NOy, respectively. The 1-second, 1-sigma detection limits are 20 pptv, 30 pptv, and 20 pptv for NO, NO2, and NOy, respectively (https://discover-aq.larc.nasa.gov/pdf/2010STM/Weinheimer20101005_DISCOVERAQ_AJW.pdf). For REAM, NOy is the sum of NO, NO2, total peroxyacyl nitrates (∑PNs), total alkyl nitrates (∑ANs) (include alkyl nitrates and hydroxyalkyl nitrates), HNO3, HONO, 2 × N2O5, HNO4, first generation C5 carbonyl nitrate (nighttime isoprene nitrate ISN1: $C_5H_8NO_4$), 2 × C5 dihydroxydinitrate (DHDN: $C_5H_{10}O_8N_2$), methyl peroxy nitrate (MPN: $CH_3O_2NO_2$), propanone nitrate (PROPNN: $CH_3C(=O)CH_2ONO_2$), nitrate from methyl vinyl ketone (MVKN: $HOCH_2CH(ONO_2)C(=O)CH_3$), nitrate from methacrolein (MARCN: $HOCH_2C(ONO_2)(CH_3)CHO$), and ethanol nitrate (ETHLN: $CHOCH_2ONO_2$).

[2] For P-3B, the concentrations of NO2_LIF, ∑PNs, ∑ANs, and HNO3 were measured by applying the thermal dissociation-laser induced fluorescence (TD-LIF) technique. The accuracy of TD-LIF measurements of NO2, ∑PNs, ∑ANs, and HNO3 is better than 15%, and the detection limit for the sum of NO2, ∑PNs, ∑ANs, and HNO3 is ~ 10 ppt $10 \ s^{-1}$ (Day et al., 2002).

[3] To compare $NO_y$ concentrations from TD-LIF measurements with those from REAM, we calculate derived-NOy as the sum of NO, NO2_LIF, ∑PNs, ∑ANs, and HNO3. Only when the concentrations of all the five species are available at the same hour in the same grid cell, we can calculate derived-NOy at the given hour in the given grid cell. Therefore, in Table 1, the averaged derived-NOy values are not exactly equal to the sum of averaged NO, NO2_LIF, ∑PNs, ∑ANs, and HNO3 concentrations that only depend on the availability of a single species. In addition, the measurement times and frequencies between NOy and derived-NOy are not the same. A comparison between these two types of data needs coincident sampling, as described in the main text.

[4] Mean $NO_x$ emissions over the six P-3B spiral sites are close (relative difference <4%) between the 36-km and 4-km REAM (Table S1).

[5] Due to different sampling times and locations between weekdays and weekends, we do not recommend a direct comparison between weekday and weekend values here.





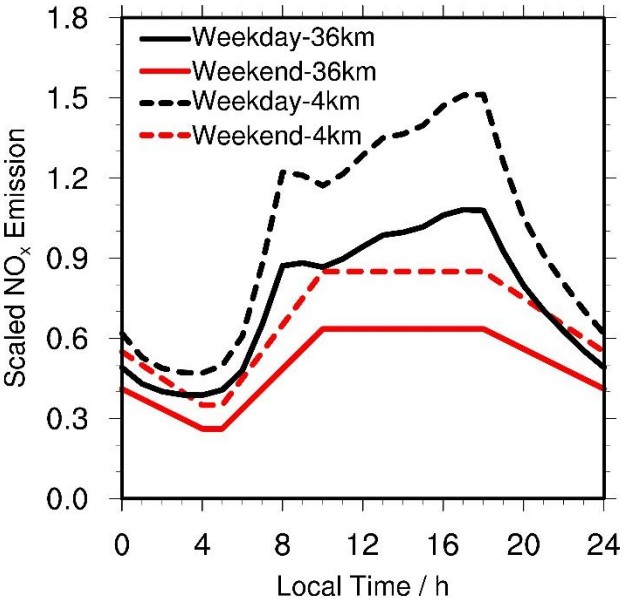

**Figure 1.** Relative diurnal profiles of weekday and weekend $NO_x$ emissions (molecules $km^{-2}$ $s^{-1}$) in the DISCOVER-AQ 2011 region (the 36/4 km grid cells over the 11 inland Pandora sites shown in Figure S1) for the 36-km and 4-km REAM. All the profiles are scaled by the 4-km weekday emission average value (molecules $km^{-2}$ $s^{-1}$).

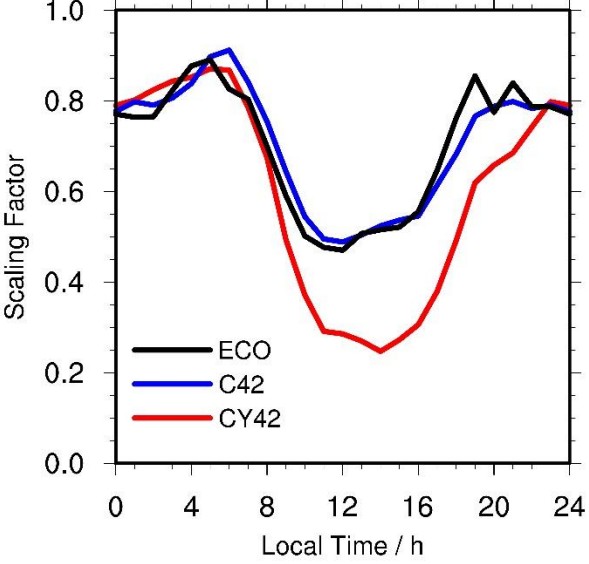

**Figure 2.** Hourly ratios of $NO_2$ measurements from the Teledyne API model 200 eup photolytic $NO_x$ analyzer to $NO_2$ from coincident catalytic instruments for 2011 July. "CY42" denotes the ratios of photolytic $NO_2$ to $NO_2$ from the Thermo Electron 42C-Y $NO_y$ analyzer in Edgewood, "C42" denotes the ratios of photolytic $NO_2$ to $NO_2$ from the Thermo Model 42C $NO_x$ analyzer in Padonia, and "ECO" denotes the ratios of photolytic $NO_2$ to $NO_2$ from the Ecotech Model 9841 T-$NO_y$ analyzer in Padonia. "ECO" ratios are also used to scale $NO_2$ measurements from the Ecotech Model 9843 T-$NO_y$ analyzer. Thermo Model 42I-Y $NO_y$ analyzer was used only in Padonia, where photolytic measurements were available, so we do not use the Thermo Model 42I-Y $NO_y$ analyzer measurements in this study.





**Figure 3.** Diurnal cycles of surface (a, c) $NO_2$ and (b, d) $O_3$ concentrations on (a, b) weekdays and (c, d) weekends during
the DISCOVER-AQ campaign in the DISCOVER-AQ region (the 36-km grid cells over the 11 inland Pandora sites shown
in Figure S1). Black lines denote the mean observations from all the 11 $NO_2$ surface monitoring sites and 19 $O_3$ surface sites
during the campaign (Figure S1), as mentioned in Section 2.5. "REAM-raw" (green lines) denotes the coincident 36-km
REAM simulation results with WRF simulated $k_{zz}$ data, and "REAM-kzz" (red lines) is the coincident 36-km REAM
simulation results with updated $k_{zz}$ data. See the main text for details. Vertical bars denote corresponding standard deviations.





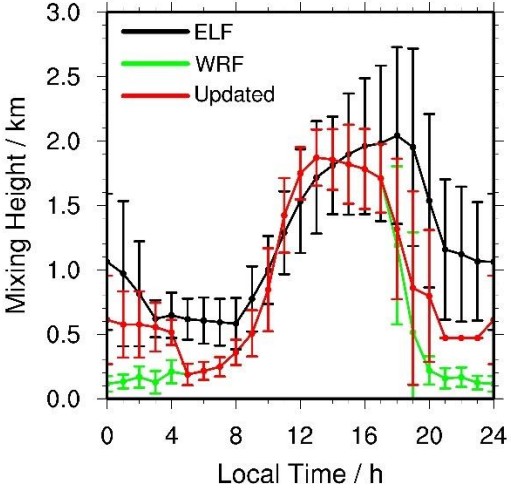

**Figure 4.** ELF observed and model simulated diurnal variations of PBLH at the UMBC site during the Discover-AQ campaign. "ELF" denotes ELF derived PBLHs by using the covariance wavelet transform method. "WRF" denotes the 36-km WRF $k_{zz}$-determined PBLHs, and "Updated" denotes updated $k_{zz}$-determined PBLHs. See the main text for details. Vertical bars denote standard deviations.



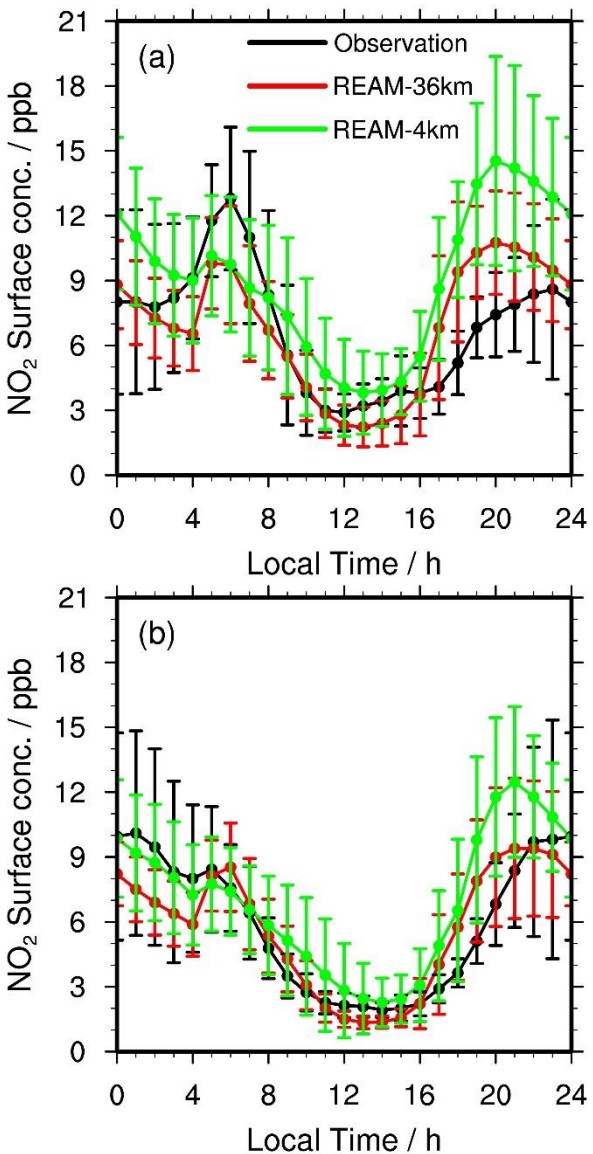

**Figure 5.** Diurnal cycles of observed and simulated average surface $NO_2$ concentrations over Padonia, Oldtown, Essex, Edgewood, Beltsville, and Aldino (Table S1) on (a) weekdays and (b) weekends. Black lines denote mean observations from the six sites. Red lines denote coincident 36-km REAM simulation results, and green lines are for coincident 4-km REAM simulation results. Error bars denote standard deviations.

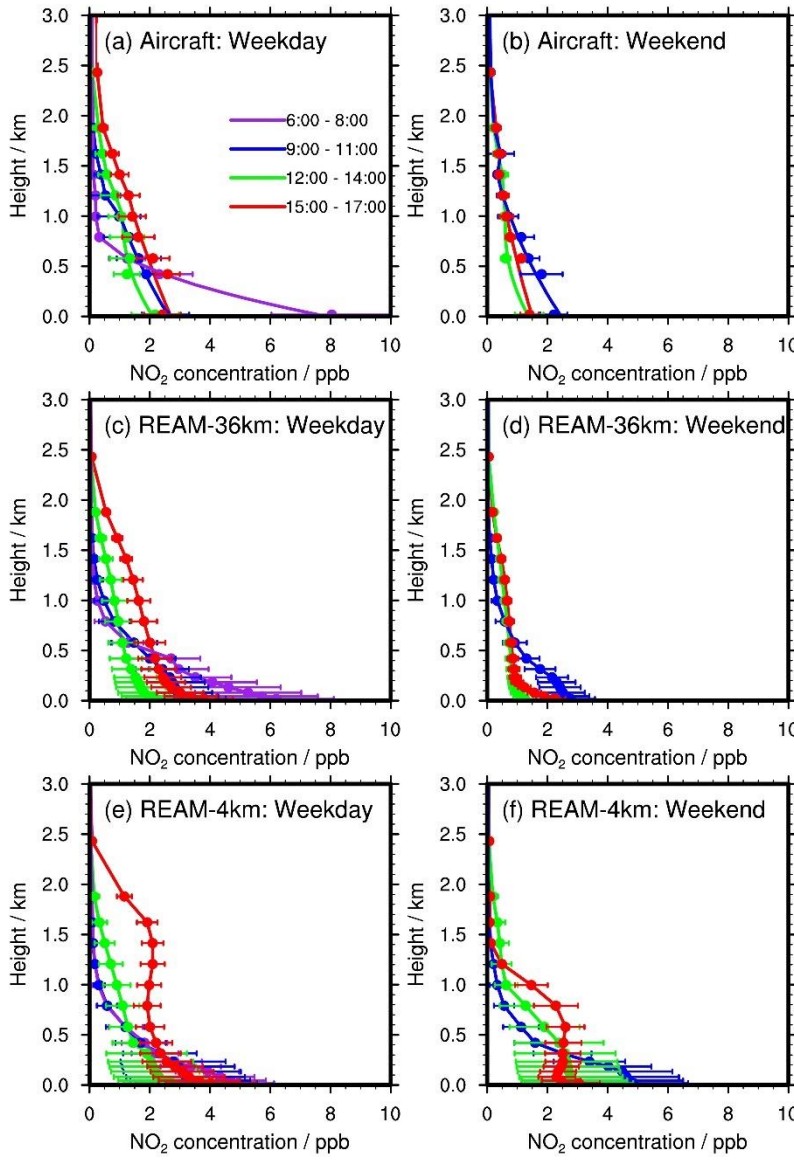

**Figure 6.** Temporal evolutions of NO$_2$ vertical profiles below 3 km on (a, c, e) weekdays and (b, d, f) weekends from the (a, b) P-3B aircraft and (c, d) 36-km and (e, f) 4-km REAM during the DISCOVER-AQ campaign. Horizontal bars denote the corresponding standard deviations. In (a) and (b), dots denote aircraft measurements, while lines below 1 km are based on quadratic polynomial fitting, as described in section 2.6. The fitting values are in reasonable agreement with the aircraft and surface measurements in the boundary layer. On weekends, no aircraft observations were made at 6:00 – 8:00 LT, and therefore no corresponding model profiles are shown.



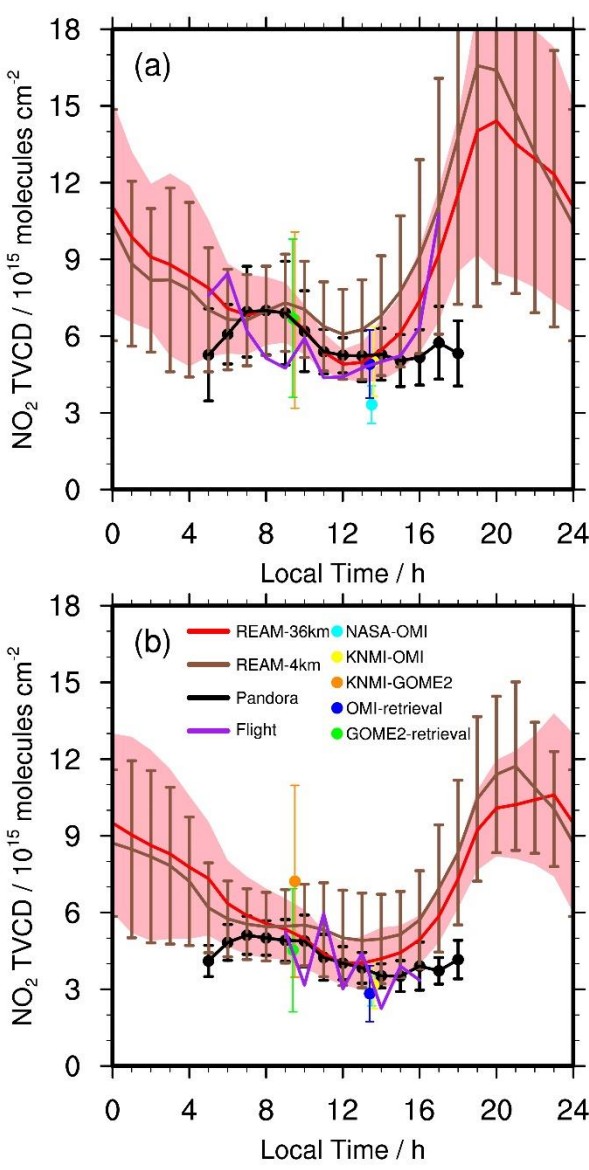

**Figure 7.** Daily variations of $NO_2$ TVCDs on (a) weekdays and (b) weekends during the DISCOVER-AQ campaign. "REAM-36km" refers to the 36-km REAM simulation results over the 11 inland Pandora sites. "REAM-4km" refers to the 4-km REAM simulation results over the 11 inland Pandora sites. "Pandora" refers to updated Pandora TVCD data. "Flight" denotes P-3B aircraft-derived $NO_2$ VCDs below 3.63 km. "NASA-OMI" denotes the OMI $NO_2$ TVCDs retrieved by NASA over the Pandora sites; "KNMI-OMI" denotes the OMI $NO_2$ TVCDs from KNMI; "KNMI-GOME2" is the GOME-2A $NO_2$ TVCDs from KNMI. "OMI-retrieval" and "GOME2-retrieval" denote OMI and GOME-2A TVCDs retrieved by using the KNMI DOMINO algorithm with corresponding 36-km REAM vertical profiles, respectively.



**Figure 8.** Weekday hourly variations of NO$_2$ VCDs at different height (AGL) bins ($<$ 3.63 km AGL, $<$ 400 m AGL, and 400
m $\sim$ 3.63 km AGL) based on P-3B aircraft-derived datasets and the 36-km and 4-km REAM results. "Flight" denotes P-3B
aircraft-derived NO$_2$ VCDs, "REAM-36km" denotes coincident 36-km REAM simulated VCDs, and "REAM-4km" denotes
coincident 4-km REAM simulated VCDs.

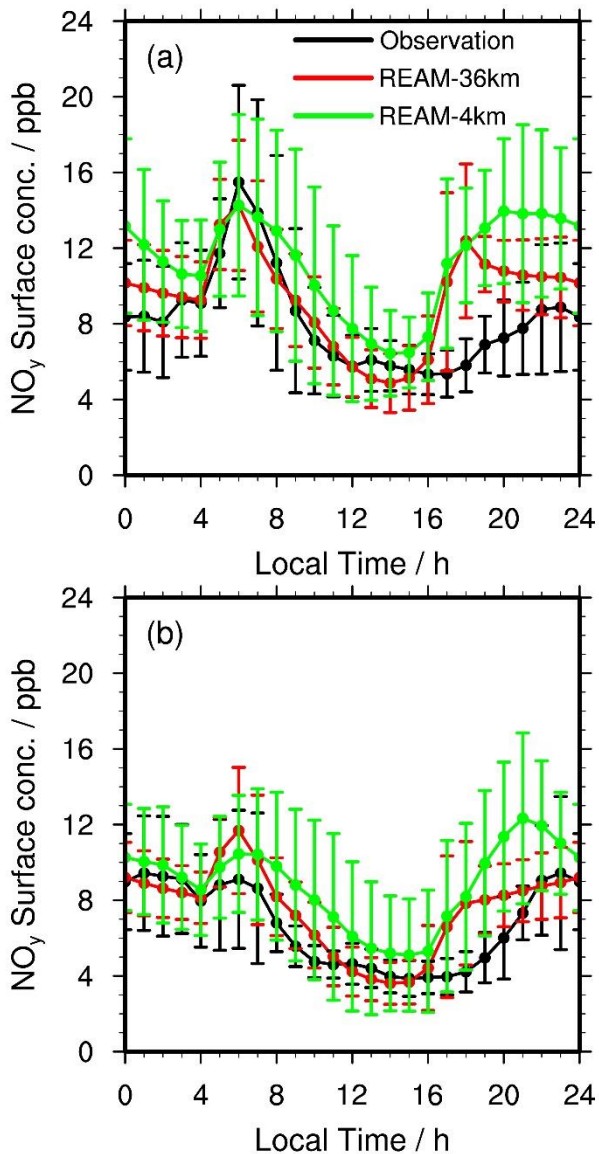

**Figure 9.** Diurnal cycles of observed and simulated average surface $NO_y$ concentrations at Padonia, Edgewood, Beltsville, and Aldino on (a) weekdays and (b) weekends. Vertical bars denote the corresponding standard deviations. It is noteworthy that the mean $NO_x$ emissions over Padonia, Edgewood, Beltsville, and Aldino are 99% higher in the 4-km than the 36-km REAM simulations (Table S1 and Figure S1).





**Figure 10.** Distributions of the scaled mean (a) ACAM $NO_2$ VCDs below the UC-12 aircraft and (b) coincident 4-km
REAM simulation results on weekdays in July 2011. (c), the distribution of the scaled NEI2011 $NO_x$ emissions on
weekdays. (d) is the scatter plot of the scaled ACAM and 4-km REAM $NO_2$ VCDs from (a) and (b). Here, we scale all
values (VCDs and $NO_x$ emissions) based on their corresponding domain averages. The domain averages of ACAM and
coincident 4-km REAM $NO_2$ VCDs are $4.7 \pm 2.0$ and $4.5 \pm 3.2 \times 10^{15}$ molecules $cm^{-2}$, respectively.

1233