# Peer review of "Comprehensive evaluations of diurnal NO2 measurements during DISCOVER-AQ 2011: Effects of resolution dependent representation of NOx emissions"

_Atmospheric Chemistry and Physics, 2020_

## Author Comment (AC1)

**Response to Reviewer #1**

Thank you for your careful and thorough reading of this manuscript and your thoughtful comments and suggestions. Our responses follow your comments (in *Italics*).

This manuscript reports the extensive comparison of the REAM chemical transport model (CTM) simulations with the NOx and NOy observations acquired during the DISCOVER-AQ 2011 over the Baltimore and Washington-DC area. The observations include the data from surface monitors, PANDORA, P3 aircraft, ACAM, and satellites OMI and GOME-2. The model results with two spatial resolutions, 36 km and 4 km are compared in order to elucidate the impact of the resolutions on the model NOx and NOy simulations. Differences between the model and observations are discussed in details and causes for the discrepancies are suggested.

The manuscript reflects the extensive works dealing with almost all available data sets to evaluate NO2 measurements and CTM results over the Baltimore and Washington-DC area for July 2011. I appreciate the efforts the authors made for this study. The manuscript will be more valuable if quality of presentation and interpretation of the results are enhanced.

**Reply:**

Thank you for your suggestions and comments. We have made several major revisions to the manuscript based on the suggestions by you and another reviewer.

- We have updated our WRF and REAM simulations using WSM6 (WRF Single-Moment 6class scheme) instead of WSM3, as listed in Table S2 (Line 34) in the revised supplemental table file. We also have downloaded the updated DISCOVER-AQ 2011 P-3B observations. All relevant results, including figures and tables, have been updated accordingly. The new results are almost the same as before except for some minor differences. The changes have no impact on our conclusions. The WSM3 results are now used as a sensitivity test (Lines 626 – 632) to confirm the reliability of our results and conclusions.
- 2) We have deleted the discussion on the reliability of 36-km NOx emissions and total NOx emission amount in the DISCOVER-AQ region but added an analysis of the 36-km NOx emission distribution issue. So the current manuscript just focuses on the distribution issue but does not include any judgment on the total NOx emission amount. Please see Lines 43 47, 125 126, 666, 718 730, 732 733, 753 789, 794, 797 806, 850 857, and 1385 1390 (Figure 15) in the revised main manuscript and Lines 281 288 (Figure S27) in the revised supplemental figure file.
- 3) We have added individual site comparisons in the supplemental figure file (Figures S19 S23, Lines 242 260) to demonstrate the NOx emission distribution issue. Please see Lines 37, 40 43, 668 678, 814, 829 831, and 837 848 in the revised main manuscript.
- We have added some more detailed explanations for the Pandora issue in the late afternoon and early morning. Please see Lines 536 – 544 in the revised main manuscript and Figure S13 (Lines 185 – 190) in the revised supplemental figure file.
- 5) We have moved the evaluation of WRF meteorological fields to a new section 3.1 and added the evaluation of vertical profiles for several meteorological variables in the new Figure S6. Please see Lines 158 183 and 339 366 in the revised main manuscript and Figure S4 S8 (Lines 108 146) in the revised supplemental figure file.
- 6) We have used stricter and more consistent criteria to filter out invalid satellite NO2 TVCDs (Lines 205 206 and 224 225). The GOME-2A morning high bias is gone, as shown in

Figure 10 in the revised main manuscript (Lines 1344 - 1353). Relevant changes in the text are in Lines 564 - 567, 818, and 823 - 826.

**Detailed responses are as follows.**

The main focus of the paper seems to be the comparison of the model simulations with the 36 km and 4 km resolution and advocate the use of 36 km in the end. I think the authors should focus more on the analysis of 4 km resolution results and causes for the similarities and discrepancies with various observations. The emissions at 36 km resolution are simply accumulations of the emissions at 4 km. It is not important to compare the emissions at the two resolutions and judge which one is better. The authors have the best spatial resolution of emission inventory data and the model simulations at the comparable scale (4 km). If the model overestimates the NOx, NOy observations at one height or vertically column integrated, that simply means the model emissions are overestimated. For the pollution hot spots in the domain, the model values are higher than the observations (judging from the ACAM data). This may be about the spatial location error in the NEI as the authors jumped to the conclusions, but it is more probable that the uncertainties in the emission factors (or activities) over populated urban or roads as represented as MOVES caused the problem. Section 3.7 should be deleted or rewritten. This section is confusing and misleading.

**Reply:**

Thank you for your suggestions and comments. As we stated in the original manuscript, the model results between 4 and 36 km resolutions are different in comparison to the observations. Modeling with a higher spatial resolution does not necessarily improve model simulations. In any research that finds superior modeling with a higher resolution, the change of resolution is usually a minor reason; the better representation of physical and dynamical processes at a higher resolution is usually more important in atmospheric models. Therefore, it is scientifically important to compare model simulations in two or more resolutions if possible, as we did in this study. We did not "advocate" the use of a lower resolution model. What we suggested is that the 4-km emission distribution of NOx emissions causes model errors in our evaluation using DISCOVER-AQ measurements, and it needs to be improved. We believe that we are in agreement with the reviewer on the importance of improving high-resolution emission inventories for NOx and other pollutants. For users of the emission inventories, errors in MOVES are part of the distribution errors in the NEI. Very few ACP readers understand the details of MOVES. To identify the issues and uncertainties in MOVES, one will have to write another paper. The reviewer appeared to misunderstand our intentions.

This paper is not meant to promote lower-resolution air quality modeling. We want to understand the reasons why the high-resolution model does not reproduce the observations. To illustrate the potential NOx emission distribution issue between 36-km and 4-km resolutions, we took a step by step approach. We first discussed the reliability and possible limitations of the 4km REAM (sections 3.1 - 3.5) and then identified NOx emissions and gradients as a major factor causing the discrepancies among the 36-km REAM, the 4-km REAM, and observations through comprehensive evaluations and diagnostics of NOx related chemistry and physics (sections 3.6). Next, through individual site comparisons, we found that the performances of the 36- and 4-km REAM simulations depend upon the observation locations. A uniform underestimation or overestimation of NOx emissions cannot explain all the model biases, and there may be some distribution biases for the NEI NOx emissions. Finally, we verified the potential distribution biases of NEI NOx emissions at both 36- and 4-km resolutions by comparing NO2 VCD distributions from OMI, GOME-2A, and ACAM with those from 36- and 4-km REAM simulations. This structure aims to make the manuscript reasonable and understandable.

In the old manuscript, we only discussed the distribution bias of NEI2011 NOx emission distributions at high resolution, but it doesn't mean that we advocated using 36 km. What we really wanted to say was that the total NOx emissions might be reasonable in the DISCOVER-AQ 2011 region since we defined the DISCOVER-AQ 2011 region as the six selected 36-km grid cells. And we also emphasized that our conclusion on the total NOx emissions was only valid in the DISCOVER-AQ 2011 region, considering the significant spatial heterogeneity of NOx emissions. In the new manuscript, we have deleted our judgments and discussions on the total NOx emissions due to the reviewer's strong objection and the relatively small size of the DISCOVER-AQ region. We have added some discussion on the NOx emission distribution issue at 36-km resolution to make the paper more balanced, which we think is what the reviewer asked for. The distribution issue is the most critical point we want to emphasize in this study.

The 36-km NEI emissions are indeed the sum of the 4-km NEI emissions. But the results from the 4-km REAM can differ significantly from the 36-km run because of nonlinear processes such as oxidation chemistry. Without comparisons between 36-km and 4-km REAM simulations, we would not know whether an issue only exists in 4-km REAM or also in coarse resolutions. Moreover, through the comparison between 36- and 4-km REAM simulations, we can derive the effects of NOx emissions and gradient on reproducing NO2 and NOy observations. It would be hard to do by only using the 4-km REAM simulation results, as NOx gradient-associated transport is hard to exclude in one simulation.

If the observations are limited, a distribution issue may be thought to be a simpler problem of overestimating or underestimating the sum of NOx emissions. It is a reason why we used as many observations as possible in the DISCOVER-AQ experiment. The intensive field measurements make it possible to analyze the issue more in depth. Please also note some previous studies only focus on polluted urban areas, while the areas of study are much broader in this work. We have used more comprehensive observations than previous studies to show the distribution bias through our diagnostics and analyses. Even in the high-emission pixels (NOx emissions > domain mean) in ACAM, as shown in Figure R1b, overestimation and underestimation of NO2 VCDs coexist. The new content we have added in the revised manuscript, such as individual site comparisons and distribution uncertainties at 36-km resolution, can further show the potential distribution bias of the NEI NOx emissions.

We have had working experience with MOVES. The setup of MOVES is very complex and involved a lot of observations and variables. It is not just a story of emission factors or Vehicle Miles Traveled (VMT or activity). Vehicle population, ages, and types (e.g, passenger cars, passenger trucks, short-haul trucks, long-haul trucks, etc.), road types, speed distributions, etc., are all crucial variables estimating running-exhaust emissions. Not all counties provided these data to the NEI setup, where MOVES may use national defaults as inputs and, therefore, missing local characteristics. Even for those counties providing local data, it is almost impossible to make a reasonable estimate of speed distributions (and some other parameters) for different vehicle types. Not to mention large uncertainties in further allocations to 4-km grids by SMOKE (MOVES can only resolve county-scale emissions). Recent GPS-based data have shown that speed distribution in the MOVES database is not accurate and cannot represent local conditions (DenBleyker et al., 2017). We understand some tunnel and roadside experiments found the overestimation of NOx emissions by MOVES for some vehicle types in some regions. But the

underestimation of NOx emissions for some vehicle types by MOVES was also found in other areas (https://www.epa.gov/sites/production/files/2017-11/documents/light\_duty\_nox.pdf). It is noteworthy that MOVES default emission factors are also based on the observations. Instead of talking about the overestimation or underestimation of MOVES NOx emission factors, we would like to think of the problem from the perspective of representativeness. Is current MOVES input data able to represent all local conditions over the US? Are the allocations of VMT, speed distribution, etc., accurate enough for different counties or high-resolution grids? Here we have to say that spatial distribution is a much more complex problem than the simplification of an overestimation or underestimation of total NOx emissions. Accurate spatial distribution requires a reasonable estimate of NOx emissions for each grid cell. Having said all these, we acknowledge the tremendous effort that went into the development of MOVES and similar programs and that they have been and continue being indispensable for air quality research.

Some readers may think negatively about the NEI  $NO_x$  inventory because of the issues we find. However, most scientists we trust understand the important contributions of the NEI  $NO_x$  emissions and our intention that identifying problems is the first step to improve our understanding and modeling capability.

---

## Author Comment (AC2)

**Response to Reviewer #2**

Thank you for your careful and thorough reading of this manuscript and your thoughtful comments and suggestions. Our responses follow your comments (in *Italics*).

*General comments:*

*The manuscript by Li et al. presents an important evaluation research work of $NO_2$ diurnal variation using observations and modelled results from DISCOVER-AQ 2011. The research topic is important and interesting to atmospheric modelling and observation communities. The approach used is comprehensive. Some of the findings (e.g., potential spatial distribution bias in emission inventory, potential bias in ground-based remote sensing instruments) in this work are important for not just modelling groups but also observation groups. But, the presentation of this work should be improved. I would recommend publishing this work if the following concerns and comments can be addressed.*

Thank you for your suggestions and comments. We have made several major revisions to the manuscript based on the suggestions by you and another reviewer.

1) We have updated our WRF and REAM simulations using WSM6 (WRF Single-Moment 6-class scheme) instead of WSM3, as listed in Table S2 (Line 34) in the revised supplemental table file. We also have downloaded the updated DISCOVER-AQ 2011 P-3B observations. All relevant results, including figures and tables, have been updated accordingly. The new results are almost the same as before except for some minor differences. The changes have no impact on our conclusions. The WSM3 results are now used as a sensitivity test (Lines 626 – 632) to confirm the reliability of our results and conclusions.

2) We have deleted the discussion on the reliability of 36-km $NO_x$ emissions and total $NO_x$ emission amount in the DISCOVER-AQ region but added an analysis of the 36-km $NO_x$ emission distribution issue. So the current manuscript just focuses on the distribution issue but does not include any judgment on the total $NO_x$ emission amount. Please see Lines 43 – 47, 125 – 126, 666, 718 – 730, 732 – 733, 753 – 789, 794, 797 – 806, 850 – 857, and 1385 – 1390 (Figure 15) in the revised main manuscript and Lines 281 – 288 (Figure S27) in the revised supplemental figure file.

3) We have added individual site comparisons in the supplemental figure file (Figures S19 – S23, Lines 242 – 260) to demonstrate the $NO_x$ emission distribution issue. Please see Lines 37, 40 – 43, 668 – 678, 814, 829 – 831, and 837 – 848 in the revised main manuscript.

4) We have added some more detailed explanations for the Pandora issue in the late afternoon and early morning. Please see Lines 536 – 544 in the revised main manuscript and Figure S13 (Lines 185 – 190) in the revised supplemental figure file.

5) We have moved the evaluation of WRF meteorological fields to a new section 3.1 and added the evaluation of vertical profiles for several meteorological variables in the new Figure S6. Please see Lines 158 – 183 and 339 – 366 in the revised main manuscript and Figure S4 – S8 (Lines 108 – 146) in the revised supplemental figure file.

6) We have used stricter and more consistent criteria to filter out invalid satellite $NO_2$ TVCDs (Lines 205 – 206 and 224 – 225). The GOME-2A morning high bias is gone, as shown in Figure 10 in the revised main manuscript (Lines 1344 – 1353). Relevant changes in the text are in Lines 564 – 567, 818, and 823 – 826.

Detailed responses are as follows.

*Specific comments:*

*L87-88. Many previous works were not properly cited. As I know, various research work has been done to convert Pandora $NO_2$ VCD to TVCD or surface values to study diurnal variations. The authors should update relevant knowledge on these. E.g., Kollonige et al., 2017; Spinei et al., 2014; Zhao et al., 2019. I believe some of the results in this work could be compared with previous findings and may cast some light on the research topic.*

**Reply:**

Thank you for your suggestion. We have added three more citations using both Pandora VCD and in-situ surface observations to investigate $NO_2$ diurnal variations. Please see Lines 88 – 93 in the revised main manuscript. Zhao et al. (2019) had two figures showing the diurnal cycles of $NO_2$ surface concentrations (model, in-situ, and Pandora-derived), so we have also cited it in Lines 400 – 401. The study of Spinei et al. (2014) was not much related to tropospheric $NO_2$ diurnal variations. It was cited in Line 247 to show the stratospheric $NO_2$ VCD diurnal variations.

*L151-176. These detailed discussions of the wind-filed and precipitations should not be done here, as the reader does not know anything about your trace gas simulation results/discrepancy yet. Such detailed discussions (the author used six figures in total, Figs. S2-S7) of potential causes should be included in a separate discussion section.*

**Reply:**

Thank you for your suggestion. As mentioned above, we have moved the evaluation of WRF meteorology to section 3.1. Please see Lines 158 – 183 and 339 – 366 in the revised main manuscript.

*L203 and L213. 36-km REAM profiles were used to calculate AMFs for both OMI and GOME-2A. Are these new AMFs have higher or lower (or comparable) resolution compared to the original AMFs used in the satellite data products? Please provide a brief description of how the model output has been smoothed or interpolated to OMI and GOME-2A grids.*

**Reply:**

Thank you for your suggestion. We first regridded corresponding 36-km REAM $NO_2$ vertical profiles to OMI/GOME-2A pixels, then calculated AMFs by using the regridded vertical profiles. Therefore, the updated AMFs have the same resolutions as the original ones from the DOMINO/GOME-2A products. We have added a brief introduction of our retrieval method in Lines 214 – 216 in the revised main manuscript.

For the regridding approach, we would like to show the details below but not in the main manuscript, as it is not easy to explain it clearly in 1-2 sentences and not directly relevant to the topic of the study either.

We first construct a latitude-longitude matrix with a resolution of 0.01° (~ 1 km), as shown by the dash lines in Figure R1. We calculate the location of each point of the matrix in the 36-km REAM domain. For example, the red, green, and purple points in Figure R1 are corresponding to (i = 50, j = 60), (i = 50, j = 60), and (i = 51, j = 59) of the 36-km REAM domain. For any

given OMI/GOME-2A pixel (determined by the corner latitudes and longitudes), as shown by the black box in Figure R1, we can obtain the matrix points located inside the satellite pixel. We know its corresponding location in the 36-km REAM domain and then the corresponding REAM $NO_2$ vertical profile for each point inside the pixel. We then average the corresponding 36-km REAM $NO_2$ vertical profiles of all points inside the pixel, which is the a priori $NO_2$ vertical profile for that satellite pixel. The updated a priori $NO_2$ vertical profile is then used to calculate AMF and $NO_2$ TVCDs. This differential-like approach can be used to satellite pixels at any scale and in any shape, as long as the latitude-longitude matrix resolution is high enough compared to those satellite pixels so that the computation error is ignorable. For OMI and GOME-2A pixels with nadir-resolutions of $13 \times 24$ km$^2$ and $80 \times 40$ km$^2$, 0.01° is enough.

[Figure]

Figure R1. Schematic of the regridding approach. The dash lines denote a 0.1° × 0.1° latitude-longitude matrix, and the black box denotes a satellite pixel. Colored points represent those 0.1° × 0.1° points inside the satellite pixel.

*L328-353. I saw at least three names for Kzz modelling, and I do have difficulty understanding which one is which. After reading this section back and forth several times, I think two Kzz modellings were used, i.e., Kzz-WRF and Kzz-modified. But, I am not sure if this Kzz-WRF is the same as Kzz-YSU. I can understand the logic of why the authors want to modify Kzz for*

*nighttime, but please improve the descriptions to make it easier for a reader to absorb your idea.*

**Reply:**

Thank you for your suggestion. Yes, there are two types of $k_{zz}$ used here: one is from the WRF simulation with the YSU scheme, and the other one is that we modify. $K_{zz}$-WRF is the same as $k_{zz}$-YSU, since our WRF simulations use the YSU scheme. We now use a consistent name — WRF-YSU — to denote the $k_{zz}$ data simulated by WRF with the YSU scheme. Please see Lines 370, 373 – 374, 378, 382 – 383, 385, 392, 1311, and 1316 – 1317, and Figure 6 in Line 1314 for relevant modifications. We hope it is easier to understand now.

*L347-351. Some justifications for the selected parameters are missing. A sensitivity test or correlation studies are needed to justify this 5 m s⁻². The idea of a magic number is not impressive. It is difficult to justify the selection with Figure 4, which shows even the modified results still have large discrepancy compare to observations.*

**Reply:**

Thank you for your suggestion. As emphasized in the manuscript, the assigned value of 5 m s$^{-2}$ is arbitrary. We did not choose a magic number. You may have noticed that we also need to select a height value and a time range for the $k_{zz}$ adjustment. In the manuscript, we used 500 m and 18:00 – 5:00 LT. Using these values is just to simplify the modification but not to best match the observations. That's why modified PBLHs are still lower than observations in the nighttime and late afternoon.

We made many sensitivity tests when finalizing the selection of 5 m s$^{-2}$, 500 m, and 18:00 – 5:00 LT in the manuscript since the beginning of this study. We have shown the sensitivity test results with $k_{zz}$ adjusted to 2 m s$^{-2}$ and 10 m s$^{-2}$ in Figure S10 in Lines 152 – 155 in the revised supplemental file and cited it in Line 396 in the revised main manuscript. Nighttime surface $NO_2$ and $O_3$ concentrations are very sensitive to $k_{zz}$. Using 2 m s$^{-2}$ also makes significant changes to the simulated results.

In fact, it is almost impossible to make some simple adjustments of $k_{zz}$ to perfectly match the vertical profile of $k_{zz}$ and diurnal variations of PBLH. As shown in Figure R2, WRF-YSU $k_{zz}$ shows a "C" shape. From afternoon to nighttime, the $k_{zz}$ values change, and the "C" shape height varies. In other words, $k_{zz}$ at different heights changes differently. We previously used a very complex equation to imitate the diurnal evolution of the $k_{zz}$ vertical profiles and try to slow down the variation rate from afternoon to nighttime, as shown below.

when $k_{zz}(t,l) \geq 0.01 \, m/s^2$,

$$k_{zz}(t+\Delta t,l) = \max\left(k_{zz}(t,l) \cdot \alpha(l)^{EF \cdot \beta(t+\Delta t)}, WRF.k_{zz}(t+\Delta t,l)\right) \qquad (R1)$$

when $k_{zz}(t,l) < 0.01 \, m/s^2$,

$$k_{zz}\left(t+\Delta t,l\right) = \max\left(k_{zz}\left(t,l\right), WRF.k_{zz}\left(t+\Delta t,l\right)\right) \tag{R2}$$

where $l$ denotes model vertical levels less than 15 ($\approx$ boundary layer top at 15:00 LT); $t$ is the current time, while $\Delta t$ is an updating time step (= 0.5 *hours*); $\alpha$ is a coefficient dependent on model levels; $\beta$ is a coefficient dependent on time; $EF$ is a coefficient related to land types, and $EF$ is 1 for urban regions and 2 for other land types; $WRF.k_{zz}$ is the original $k_{zz}$ from the WRF simulation. Equations (R1) and (R2) calculate $k_{zz}$ at the next time step with the current $k_{zz}$. The equations are only active when $t > 15:00$ LT and $t < 5:00$ LT. The updated $k_{zz}$ values are decreasing more slowly than the original WRF-YSU values since later afternoon and satisfy the vertical characteristics shown in Figure R2. The derived PBLH can match the observations in Figure 6 very well. The equations also consider the effect of land use/land cover. However, these equations have no physical meaning and are inappropriate to be used in the manuscript.

Anyway, 5 m s$^{-2}$ is not a magic value but just to mitigate the nighttime vertical mixing problem. The selection of 5 m s$^{-2}$ is not intended to and neither able to completely solve the nighttime vertical mixing bias. Not to mention the site differences. Readers are free to make their own adjustments in their studies if nighttime mixing is underestimated.

[Figure]

Figure R2. Vertical profiles of WRF-YSU simulated $k_{zz}$ at different local times in July 2011 at the UMBC site.

*L364-369. I am worried that the ground observations from various sites should not be studied as a single group. Different local emissions patterns should be addressed. E.g., do all 11 NO$_2$ sites show the same concentration peak values at 5:00-6:00 LT? Do we see any differences between rural and urban sites?*

**Reply:**

We understand your concern about different local emissions patterns at different sites. EPA indeed shows that in some rural regions, the $NO_x$ emissions show a unimodal diurnal pattern with a peak around noontime. However, the 11 $NO_2$ sites in Figure 5 are based on 36-km REAM. On the one hand, the 36-km REAM cannot resolve urban and rural well. On the other hand, all the 11 sites were not so rural, as they are all located around the Baltimore-Washington urban regions. Their emissions are still high and have similar emission diurnal variations as urban regions. Figure R3 shows the 36-km $NO_x$ emission diurnal cycles for each of the 11 sites in Figure 5. All the sites have similar diurnal patterns and show a sharp increase of $NO_x$ emissions in the early morning ($NO_x$ emissions may be biased due to the distribution issue of NEI2011 at 36-km resolution).

Figure R4 shows that the monthly weekday observations at all the 11 sites peak around 6:00 LT. We do not find significant differences among these sites. Our 36-km REAM with updated $k_{zz}$ cannot reproduce all the observed peaks at different sites mainly due to the remaining biased nighttime vertical mixing. However, the 36-km REAM can still somewhat capture the increase of $NO_2$ surface concentration around 6:00 LT at each site.

It is possible that $NO_2$ surface concentrations peak at other hours of the day but not around 5:00 – 6:00 LT. Figure 5 in the revised main manuscript shows two general conditions. 1) Nighttime vertical mixing is very weak, then $NO_2$ accumulates in the surface layer, possibly producing a peak around midnight, as shown by the REAM simulation with the original WRF-YSU $k_{zz}$. 2) The early morning increase of $NO_x$ emissions is mitigated or removed entirely (different $NO_x$ emission diurnal variations), leading to a much weaker or complete missed surface $NO_2$ morning peak as shown in Figure 5b. If one is concerned with a specific day at a particular site, anything can happen depending on the local conditions in the day, e.g., Thompson et al. (2019) showed that $NO_2$ surface concentration suddenly peaked around 13:00 LT in one day at an observation site in Korea (Figure 2 in their paper).

As mentioned above, we have added individual-site (the 11 sites in Table S1) comparison results in Figure S19 – S23 (Lines 242 – 260 in the revised supplemental figure file) to emphasize the $NO_x$ emission distribution issue at both 36- and 4-km resolutions.

[Figure]

Figure R3. 36-km NO$_x$ emission diurnal variations for the 11 sites in Figure 5 in the revised main manuscript. The unit is 10$^{21}$ molecules km$^{-2}$ s$^{-1}$.

[Figure]

Figure R4. Diurnal variations of observed and 36-km REAM simulated NO$_2$ surface concentrations at different sites for weekdays in July 2011. The subplot order is corresponding to the site order in Table S1 in the revised supplemental table file.

*L388-393. The general impression from Figure 5 is the REAM-4km shows a higher bias than REAM-36km compared to observations. But, this might be misleading. For example, if one looks at Figure 5b from 00:00 to 5:00 LT, the green line shows a better agreement with observations. Please provide some comments on this. The study sites should be grouped into at least two categories, e.g., rural and urban.*

**Reply:**

Thank you for your suggestion. It should be in the daytime but not at night in Line 436 in the revised main manuscript. We have corrected it. Since nighttime vertical mixing is weak, NO$_2$ is primarily concentrated in lower layers, leading to large horizontal gradients. Therefore, horizontal transport plays a crucial role in nighttime NO$_2$ concentrations and VCDs. If nighttime

vertical diffusion is not simulated well, horizontal transport can be much different. As mentioned above, since our adjustment of nighttime $k_{zz}$ is not perfect, we did not use nighttime comparisons in our evaluations of $NO_x$ emissions. The nighttime vertical mixing uncertainties have little impact on daytime $NO_2$ surface concentrations and TVCDs (Figures 5 and R5); therefore, we mainly used the daytime simulation results and observations in our analysis and discussion. The nighttime issue is mainly discussed in section 3.2 in the revised main manuscript to describe the underestimation of nighttime vertical mixing. We have added the emphasis of the "daytime" in Lines 436, 503 – 504, and 605 in the revised main manuscript.

In addition, Figure 7b (the old Figure 5b) in the revised main manuscript is for the weekend. It is noteworthy that weekend $NO_x$ emissions are scaled to two-thirds of weekday $NO_x$ emissions for all sites and have the same diurnal variations, as mentioned in section 2.1 (Lines 142 – 149) in the revised main manuscript. Therefore, potential uncertainties exist in the weekend $NO_x$ emissions. It is possible that 4-km REAM provides a more reasonable estimate of $NO_x$ emissions at night on weekends. We have no evidence showing that the rural-urban issue contributes to the comparison results during 0:00 – 5:00 LT in Figure 7b.

As mentioned above, we have added individual-site comparison results in Figure S19 – S23 (Lines 242 – 260 in the revised supplemental figure file) to emphasize the $NO_x$ emission distribution issue at both 36- and 4-km resolutions.

[Figure]

Figure R5. Comparisons of NO$_2$ TVCD diurnal variations between two 36-km REAM simulations on (a) weekdays and (b) weekends for July 2011. "REAM-raw" denotes the 36-km REAM simulation results with WRF-YSU simulated $k_{zz}$ data, and "REAM-kzz" is the 36-km REAM simulation results with updated $k_{zz}$ data.

*L435-442 and Fig. S14. I guess the authors want to show the Pandora TVCD should be corrected; otherwise, the results could be biased low due to a missing surface layer. I agree with the assumption, but it needs to be studied carefully (Fig. S14 shows some indication but not good enough). Fig. S14a shows that for some sites (e.g., SERC), one can expect Pandora to miss up to 20% of NO₂ columns. However, this is not reflected by Fig. S14b at all. If this 20% difference is true, it can be verified relatively easier than other sites. Could you plot Fig. S14b for each Pandora site separately?*

**Reply:**

Thank you for your comments. We used the averages of all 11 sites in Figure S12b (the old Figure S14b) in the revised supplemental figure file, so the differences between scaled and unscaled Pandora TVCDs are not so large (the relative difference can reach up to 6% around 6:00 LT). Figure R6 shows the difference for each Pandora site. Except for the four sites (UMCP, UMBC, SERC, and GSFC) significantly above the ground surface, all other sites have almost the same scaled and unscaled $NO_2$ TVCDs. For SERC, at 6:00 LT, the scaled $NO_2$ TVCDs is ~$5 \times 10^{15}$ molecules $cm^{-2}$, about 25% higher than the unscaled value (~$4 \times 10^{15}$ molecules $cm^{-2}$), consistent with Figure S12a. Since Figure S12a shows the same result as Figure R6 (the scaling ratios are the same), we don't think it is necessary to include Figure R6 in the manuscript.

The scaling may be useful for individual site comparison if the site is significantly above the ground surface. At the beginning of this study, we hoped that the scaling effect could be used to explain the Pandora's distinct diurnal variations from other datasets in the early morning and late afternoon. However, it cannot do that.

[Figure]

Figure R6. Same as Figure S12b in the revised supplemental figure file but for individual Pandora sites.

*L469-488. The findings here are critical for the research community to understand the discrepancy between aircraft, ground-based in situ, ground-based remote sensing, and models. The synthetic aircraft TVCDs have better agreement with REAM especially for 15:00 to 17:00 LT. The agreements between REAM and aircraft profiles (Figure 6) are very nice. So, for me, it looks like Pandora TVCDs are the one that has a major low bias. But, Figure 5 also shows that the REAM has a large positive bias compared to ground-based in situ observations from 15:00 to 17:00 LT (especially for REAM-4km). Can authors conclude if Pandora TVCDs are not accurate in this period? These results may affect the claim of accuracy of Pandora NO2 VCD is 2.7×10$^{15}$ molecules cm$^{-2}$ in L218. Also, from Fig. S9, it is clear that the observed diurnal variations at different sites could be very different. This matched with the large error bars on the REAM modelled results in Fig. 7. But why Pandora TVCDs from 11 sites show very stable results (small error bars) in Figure 7? The current explanations are not good enough to convince me. Besides understanding the model resolutions, this could be another highlight of this research work. So, I would suggest the authors provide more investigation, explanations, or discussions.*

**Reply:**

Thank you for your suggestions. The surface layer (1$^{st}$ layer of the REAM model) only contributes a small part of $NO_2$ TVCDs due to its shallow depth. The positive biases of $NO_2$ surface concentrations in REAM may be related to still underestimated vertical mixing in the afternoon (Figure 6 in the revised main manuscript). However, it is noteworthy that vertical mixing only affects the vertical distribution of $NO_2$ but not $NO_2$ TVCDs directly (vertical mixing can slightly affect $NO_2$ TVCDs indirectly as $NO_2$ lifetime is somewhat different at different heights). Therefore, the positive bias of $NO_2$ surface concentrations in REAM cannot provide any significant information for $NO_2$ TVCDs.

According to the model diagnostics, the sharp increase of $NO_2$ TVCDs in the late afternoon is mainly due to the sharp decrease of chemical loss (Figure 9 in the revised main manuscript). We think the model diagnostic result is reasonable. However, we cannot conclude that Pandora is inaccurate in the late afternoon. We have added more detailed explanations in Lines 536 – 544 in the revised main manuscript and Figure S13 (Lines 185 – 190) in the revised supplemental figure file. In our opinion, the most crucial point is that Pandora FOV is so small, and the instrument is located on the ground surface. Therefore, Pandora only covers a small area of air mass and can measure different air columns in the early morning, noontime, and late afternoon. Considering the significant spatial heterogeneity of $NO_2$, the measured $NO_2$ TVCDs can differ from each other significantly. In summary, Pandora measured $NO_2$ TVCDs are very different from those measured by satellite and simulated by models, especially in the early morning and late afternoon. Whether Pandora measurement can represent the average of a $36 \times 36$ km$^2$ column depends on the heterogeneity of $NO_2$ in that column. To evaluate the accuracy of Pandora, we need similar high-resolution instruments.

Figure 10 (the old Figure 7) only considers the temporal standard deviations of 21 weekdays and 10 weekend days in July 2011. We first calculated the mean hourly $NO_2$ TVCDs of the 11 Pandora sites. Considering the significant spatial heterogeneity of $NO_2$, we hoped that the average of 11 Pandora sites could represent the regional characteristics. Then, we calculated the monthly mean $NO_2$ TVCDs and corresponding standard deviations at each hour for weekdays

and weekends in July 2011. REAM results were processed in the same way. Therefore, discrepancies among different sites have not been considered in Figure 10 in the revised main manuscript. We have added individual site comparisons in Figure S23 in the revised supplemental figure file, showing the discrepancies among different Pandora sites.

*Technical corrections:*

*L194. Please modify the description of estimated uncertainty. "molecules cm$^{-2}$ + 25%" does not make sense.*

**Reply:**

Thanks. We have changed it to "an absolute component of $1.0 \times 10^{15}$ molecules cm$^{-2}$ and a relative AMF component of 25%". Please see Lines 201 – 202 in the revised main manuscript.

*L217-218. The description of the precision of Pandora NO$_2$ VCD is not correct. In Herman et al. 2009, the 0.01 DU (or $2.7 \times 10^{15}$ molecules cm$^{-2}$) precision is for slant column (not VCD). For Pandora NO$_2$ VCD, the estimated precision is about 0.02 DU (e.g., Zhao et al., 2020).*

**Reply:**

Thank you for your suggestion. Yes, according to Herman et al. (2009), the 0.01 DU precision is indeed for SCD. We have corrected it. Please see Lines 230 – 231.

*L288-294. The scale ratios look consistent between ECO and C42. The one that needs extra caution is CY42 (Thermo Model 421I-Y). But, if the Thermo Model 42I-Y NOy analyzer measurements are not used in this study at all (see L1175-1176), there is no need to include such detailed discussions (it will only confuse the reader). Or, at least, this information should be moved to supplement. I would suggest authors move other figures such as Fig. S1 to here, which should be more important (for the reader to understand the model scales/grids and locations of observations used in this study).*

**Reply:**

Thanks. We have deleted the discussion related to the Thermo Model 42I-Y NO$_y$ analyzer. Please see Lines 1302 – 1303 in the revised main manuscript. And we have moved the old Figures S1 and S2 to Figures 1 and 2 in the revised main manuscript.

*L377. Figure 6 is used before Figure 5. Please swap the order of the figures.*

Reply:

Thanks. We have deleted the references of old Figures 6 and S13 here. We have mentioned sections 3.3 and 3.4 in the revised main manuscript, so it is unnecessary to refer to the figures again. Please see Lines 422 – 423 in the revised main manuscript.

*Figure 8. Please use different symbols for >400m and <400m lines. Also, the caption said there are three bins, but I did not see proper labels for the "400m – 3.63 km". Are those >400m lines represents "400m – 3.63km" results? Please make sure the legends match with the caption.*

**Reply:**

Thank you for your suggestion. Yes, "> 400 m" means "400 m – 3.63 km". We have corrected it. Please see Figure 11 (Lines 1355 – 1361) in the revised main manuscript. And we have used 300 m to separate different height bins to match the newly downloaded P-3B observations, which can go down as low as about 300 m, as shown in Figure 8 in the revised main manuscript. The modification doesn't change the results or conclusions.

*Figure 10. Description of the purple circles on panels a-c is needed.*

**Reply:**

Thanks. We have added a sentence describing the purple circles. Please see Lines 1380 – 1381 in the revised main manuscript.

*Fig. S1 should be modified. The symbols for different observations jams together and very difficult to see. One should use other means to show instruments at a single site, e.g., a pie chart.*

**Reply:**

Thank you for your suggestion. We have changed the marker patterns for different instruments so that the figure is clearer. Please see Figure 1 in Line 1277 in the revised main manuscript.

*Reference*

*Kollonige, D. E., Thompson, A. M., Josipovic, M., Tzortziou, M., Beukes, J. P., Burger, R., Martins, D. K., Zyl, P. G. van, Vakkari, V. and Laakso, L.: OMI Satellite and Ground-Based Pandora Observations and Their Application to Surface NO$_2$ Estimations at Terrestrial and Marine Sites, J. Geophys. Res., 123(2), 1441–1459, https://doi.org/10.1002/2017JD026518, 2017.*

*Spinei, E., Cede, A., Swartz, W. H., Herman, J. and Mount, G. H.: The use of NO$_2$ absorption cross section temperature sensitivity to derive NO$_2$ profile temperature and stratospheric–tropospheric column partitioning from visible direct-sun DOAS measurements, Atmospheric Measurement Techniques, 7(12), 4299–4316, https://doi.org/10.5194/amt-7-4299-2014, 2014.*

*Zhao, X., Griffin, D., Fioletov, V., McLinden, C., Davies, J., Ogyu, A., Lee, S. C., Lupu, A., Moran, M. D., Cede, A., Tiefengraber, M. and Müller, M.: Retrieval of total column and surface NO$_2$ from Pandora zenith-sky measurements, Atmos. Chem. Phys., 19(16), 10619–10642, https://doi.org/10.5194/acp-19-10619-2019, 2019.*

*Zhao, X., Griffin, D., Fioletov, V., McLinden, C., Cede, A., Tiefengraber, M., Müller, M., Bognar, K., Strong, K., Boersma, F., Eskes, H., Davies, J., Ogyu, A. and Lee, S. C.: Assessment of the quality of TROPOMI high-spatial-resolution NO$_2$ data products in the Greater Toronto Area, Atmos. Meas. Tech., 13(4), 2131–2159, https://doi.org/10.5194/amt-13-2131-2020, 2020.*

**Reply:**

Thank you for providing the references.

**References**

Herman, J., Cede, A., Spinei, E., Mount, G., Tzortziou, M., and Abuhassan, N.: $NO_2$ column amounts from ground-based Pandora and MFDOAS spectrometers using the direct-Sun DOAS technique: Intercomparisons and application to OMI validation, J. Geophys. Res.-Atmos., 114, https://doi.org/10.1029/2009JD011848, 2009.

Spinei, E., Cede, A., Swartz, W. H., Herman, J., and Mount, G. H.: The use of $NO_2$ absorption cross section temperature sensitivity to derive $NO_2$ profile temperature and stratospheric–tropospheric column partitioning from visible direct-sun DOAS measurements, Atmos. Meas. Tech., 7, 4299-4316, https://doi.org/10.5194/amt-7-4299-2014, 2014.

Thompson, A. M., Stauffer, R. M., Boyle, T. P., Kollonige, D. E., Miyazaki, K., Tzortziou, M., Herman, J. R., Abuhassan, N., Jordan, C. E., and Lamb, B. T.: Comparison of Near‐Surface $NO_2$ Pollution With Pandora Total Column $NO_2$ During the Korea‐United States Ocean Color (KORUS OC) Campaign, J. Geophys. Res.-Atmos., 124, 13560-13575, https://doi.org/10.1029/2019JD030765, 2019.

Zhao, X., Griffin, D., Fioletov, V., McLinden, C., Davies, J., Ogyu, A., Lee, S. C., Lupu, A., Moran, M. D., Cede, A., Tiefengraber, M., and Müller, M.: Retrieval of total column and surface $NO_2$ from Pandora zenith-sky measurements, Atmos. Chem. Phys., 19, 10619-10642, https://doi.org/10.5194/acp-19-10619-2019, 2019.

---

## Author Response (AR2)

Dr. Yugo Kanaya

Editor

Atmospheric Chemistry and Physics

June 1, 2021

Dear Dr. Kanaya,

Subject: Revision of manuscript #acp-2020-1193

Thank you and the two reviewers again for reviewing our manuscript. We appreciate your thoughtful comments and suggestions. We have carefully reviewed the comments and revised the manuscript accordingly. Our responses are given in a point-by-point manner below. We have also attached a version of the manuscript and supplement with tracked changes and hope the revised manuscript is suitable for publication.

Please address all correspondence concerning this manuscript to Dr. Yuhang Wang (yuhang.wang@eas.gatech.edu). Thanks again for your time.

Sincerely,

Jianfeng Li

Atmospheric Sciences and Global Change Division

Pacific Northwest National Laboratory

Richland, Washington, US, 99354

**Response to Editor**

Thank you for your careful and thorough reading of this manuscript and your thoughtful comments and suggestions. Our responses follow your comments (in *Italics*).

*Comments to the Author:*

*Dear Authors,*

*The two reviewers found that the manuscript has been improved based on the original comments and the authors' additional analyses/revisions. However, one reviewer still suggests minor revision is necessary before publication. I would appreciate it if the authors could further consider the comments from Reviewer #2. Thank you very much.*

**Reply:**

Thank you for your comments. We have revised the manuscript based on the suggestions from the two reviewers. Besides, we have made some other slight changes to the language. Please see Lines 522 and 747 in the revised main manuscript and Lines 117 – 119 in the revised supplemental figure file.

**Response to Reviewer #1**

Thank you for your careful and thorough reading of this manuscript and your thoughtful comments and suggestions. Our responses follow your comments (in *Italics*).

*The authors have addressed most of my concerns/suggestions. I agree with the authors that night-time/early-morning PBLH is one of the difficulties the modelling community needs to study/solve. The mismatch found between ground-based remote sensing instruments and modelling results is interesting and important for air-quality researchers. I cannot say I agree with all the findings and results in the manuscripts, but I think the authors have provided enough descriptions, thoughts, and reasoning for readers and researchers. I would like to support the publication of this good-quality work on ACP.*

*Only a few minor technical corrections are left.*

**Reply:**

Thank you for your positive comments. After addressing the reviewer's suggestions, we have also updated two references from Lamsal et al. (2021) and Judd et al. (2020). Please see Lines 171, 174, 503, 981 – 982, and 1017 – 1020. We have deleted the $R^2$ values for wind direction in Line 321 because wind direction is a periodic variable which is not suited for computing a linear correlation. Figure S5 shows that our WRF simulation predicts wind direction well.

*Technical corrections:*

*Figs. 10 and 13 (revised manuscript). I expect the shading areas (e.g., red areas on Fig. 10) are also the 1 std of the data (same as the error bars). But, I would suggest the authors make this clear. Also, I did not see any reason why the KNMI-GOME2 results in the revised manuscript became "better" (in terms of agreement with other models/observations). I would expect just some outliers been removed. But, if any criteria have been changed, one should provide such details at least in the response file.*

**Reply:**

Thank you for your suggestions. Yes, the shading areas in Figures 10 and 13 show the standard deviations of 36-km REAM simulation results. We clarified the captions of Figs. 10 and 13. Please see Lines 1274 – 1276 and Lines 1296 – 1297 in the revised main manuscript.

There are two reasons for the "better" KNMI GOME-2A results. One is the removal of some outliers, as the reviewer mentioned. The other one is the inclusion of negative $NO_2$ TVCDs. In the first version of the manuscript, we only selected positive $NO_2$ TVCDs. However, according to the user guide of NASA OMNO$_2$ (v4.0) (https://aura.gesdisc.eosdis.nasa.gov/data/Aura_OMI_Level2/OMNO2.003/doc/READ

ME.OMNO2.pdf), we should consider all valid measurements, regardless of the sign, to avoid biases. Negative $NO_2$ TVCDs are produced when DOAS-derived total $NO_2$ slant column densities (SCDs) are lower than stratospheric $NO_2$ SCDs. Stratospheric $NO_2$ SCDs can be derived from model results or measurements from unpolluted areas. We want to emphasize that we used the same criteria (Lines 179 – 180 in the revised main manuscript) for all satellite $NO_2$ TVCDs, including KNMI and NASA products and our retrievals with 36-km REAM simulated $NO_2$ vertical profiles, to keep sampling consistent among these results.

*Fig. S23 (revised manuscript). Please consider modifying the subpanels to use the same y-range (e.g., 0-24). I think the authors want to show the results clearly for each site, but the scale of the difference (between model and observations) might be more important in the current stage.*

**Reply:**

Thanks. We now use the same vertical scale in Figure S23. Please see Lines 174 – 175 in the revised supplemental figure file.

**References:**

Judd, L. M., Al-Saadi, J. A., Szykman, J. J., Valin, L. C., Janz, S. J., Kowalewski, M. G., Eskes, H. J., Veefkind, J. P., Cede, A., Mueller, M., Gebetsberger, M., Swap, R., Pierce, R. B., Nowlan, C. R., Abad, G. G., Nehrir, A., and Williams, D.: Evaluating Sentinel-5P TROPOMI tropospheric $NO_2$ column densities with airborne and Pandora spectrometers near New York City and Long Island Sound, Atmos. Meas. Tech., 13, 6113-6140, https://doi.org/10.5194/amt-13-6113-2020, 2020.

Lamsal, L. N., Krotkov, N. A., Vasilkov, A., Marchenko, S., Qin, W., Yang, E. S., Fasnacht, Z., Joiner, J., Choi, S., Haffner, D., Swartz, W. H., Fisher, B., and Bucsela, E.: Ozone Monitoring Instrument (OMI) Aura nitrogen dioxide standard product version 4.0 with improved surface and cloud treatments, Atmos. Meas. Tech., 14, 455-479, https://doi.org/10.5194/amt-14-455-2021, 2021.

**Response to Reviewer #2**

Thank you for your careful and thorough reading of this manuscript and your thoughtful comments and suggestions. Our responses follow your comments (in *Italics*).

*The manuscript was majorly revised and improved. Detailed analysis in the manuscript and supporting information would be valuable for the science community in this field. However, I would like to recommend authors to rewords abstract and main text to reflect what is in the manuscript considering uncertainties in the data and analysis. Main concerns are addressed below.*

**Reply:**

Thank you for your comments and suggestions. We have added more discussion of the uncertainties/limitations of the ELF mixed-layer height and ACAM NO$_2$ VCD data. Details are as follows.

*I would like to ask the authors to inform uncertainties in mixing heights retrieved from lidar during nighttime (or stable condition). The mixing height determined by aerosol backscatter may not be a direct indicator for stable boundary layer height, but the residual layer of aerosols. Compton et al. (2013) evaluated the lidar mixing height observed only during daytime. Because ELF data are not a reliable stable boundary layer height, the updates in this manuscript should be regarded as a sensitivity test, not a correction. In this regard, abstract and main text need to be reworded carefully. More information on the location and characteristics (number of data and uncertainty) of ELF observations would be helpful. Note that nocturnal boundary layer height from original YSU scheme is within the range of boundary layer height measured and modeled (Steeneveld et al., 2007; Koracin and Berkowics, 1988; Nieuwstadt and Tennekes, 1981). In the abstract, "However, nighttime mixing in the model needs to be enhanced to reproduce the observed NO2 diurnal cycle in the model". Based on the comments above, I think this should be rephrased. Furthermore, uncertainties in the NOx emissions in nighttime were not well estimated or understood. This part needs to be mentioned.*

**Reply:**

Thank you for your comments and suggestions. We have added more explanations and discussion about the ELF data, substituted PBLH with mixed-layer height (MLH), and revised the abstract and main text accordingly. Please see Lines 38 − 40, 356 − 358, 360 − 385, 387, 390 − 394, 423, 446, 456, 595, 765, 996 − 998, 1237 − 1243.

Compton et al. (2013) indeed only evaluated the ELF mixing height with other measurements in the daytime, and the covariance wavelet transform (CWT) method is also designed for daytime (sunrise to sunset) but not nighttime. However, the residue layer (RL) issue is considered in the algorithm. When a RL is encountered, mixing

height will be generally searched for below the RL. The limitation is that they set RL = 1 km constantly in the algorithm. According to Compton et al. (2013), one major problem during nighttime is the insufficient vertical resolution of the CWT technique as nighttime mixing heights are much lower than daytime.

We agree with you that there are potential larger uncertainties for the ELF data at night than in the daytime. However, as explained in the revised main manuscript in Lines 373 – 378, Figure 6 still shows underestimated WRF-YSU MLHs compared to ELF observations in the early morning after sunrise and the late afternoon before sunset. They are daytime measurements and have been evaluated by using Radar wind profiler observations and Sigma Space mini-micropulse lidar data. Moreover, the nighttime MLHs in Figure 6 are comparable to those measured by the Vaisala CL51 ceilometer at the Chemistry And Physics of the Atmospheric Boundary Layer Experiment (CAPABLE) site in Hampton, Virginia (Knepp et al., 2017). We now use the term, MLH, following Knepp et al. (2017).

UMBC is an urban site surrounded by a mixture of constructed materials and vegetation. Maybe urbanization-associated surface roughness change, anthropogenic heat release, and heat storage are potential causes of the high nighttime MLHs. We have reworded the sentence in Lines 38 – 40 in the abstract to avoid overemphasizing the importance of $k_{zz}$ adjustment.

In the previous submission, we mentioned the potential uncertainties of nighttime $NO_x$ emissions in Lines 390 – 394 in the revised main manuscript, and our previous sensitivity tests in Figure R1 indicate that nighttime $NO_x$ emissions need to be reduced by at least 50% to match REAM results with $NO_2$ observations within uncertainties, and the reduction needs to be at least 67% to match $O_3$ concentrations between REAM and observations. Without additional robust evidence, such significant reductions are unreasonable, as we stated in Line 394 in the revised main manuscript. We have rewritten the sentences so that they are more apparent. Please see Lines 390 – 394 in the revised main manuscript.

[Figure]

Figure R1. Diurnal cycles of surface $NO_2$ (a, c) and $O_3$ (b, d) concentrations on weekdays (a, b) and weekends (c, d) during the DISCOVER-AQ campaign in the DISCOVER-AQ region. "REAM-raw" (green lines) denotes the REAM simulation with NEI2011 emissions; "REAM-75%" denotes the REAM simulation with $NO_x$ emissions from 18:00 – 5:00 LT reduced by 25%; "REAM-50%" denotes the REAM simulation with $NO_x$ emissions from 18:00 – 5:00 LT reduced by 50%; "REAM-33%" denotes the REAM simulation with $NO_x$ emissions from 18:00 – 5:00 LT reduced by 67%; "REAM-25%" denotes the REAM simulation with $NO_x$ emissions from 18:00 – 5:00 LT reduced by 75%. Black lines denote the observations during the campaign, and black vertical bars denote corresponding standard deviations.

*Nighttime biases in NO2 and NOy at the surface were much reduced when the updated YSU Kzz was used. However, column NO2 concentration in Figure 10 during nighttime are much larger than those from PANDORA. It is possible that the PANDORA*

*observations are underestimated in the morning and late afternoon, but it is also possible that the model columns are overestimated due to the emission uncertainty and this problem could not be fixed with the updated Kzz. Figure S2 exhibits dynamic changes in PANDORA NO2 columns during daytime and the model well reproduced these changes except late afternoon to nighttime (Figure S23). In the abstract, "Another discrepancy is that Pandora measured NO2 TVCDs show much less variation in the late afternoon than simulated in the model". Can it be the case that the model columns vary too much?*

**Reply:**

Thank you for your suggestions. We think it is necessary to clarify that when we talked about the discrepancies between REAM simulation results and Pandora, we didn't mean that Pandora was wrong. What we did was to describe the results and explain why they are different. As illustrated in our responses in the first round of the review process and Lines 523 – 540 in the revised main manuscript, Pandora measured different columns of air at different times of the day and is sensitive to local conditions. It is highly possible that Pandora measurements cannot represent the average conditions of a REAM grid cell, considering the potential significant spatial heterogeneity of $NO_x$, especially in the early morning and late afternoon when $NO_x$ lifetime is long, and $NO_x$ accumulation is much more apparent than mid-day. We can say that Pandora has limitations in evaluating current model results; however, it doesn't mean that Pandora is wrong, or REAM simulations are wrong because the measurements and model results are not directly comparable. The performance of REAM simulations in the afternoon was validated using P-3B aircraft derived $NO_2$ VCDs, as shown in Figure 10 in Line 1267 in the revised main manuscript. We have reworded the sentence in Lines 517 – 518 in the revised main manuscript to avoid potential misunderstanding that we meant Pandora was wrong.

*In Figure 14, the authors indicated that the purple circle denote a small region surrounded by high-NOx emission pixels and with high NO2 VCDs in the 4-km REAM but low NO2 VCDs in ACAM. But Figure S25 shows enhanced columns in the purple circle region in ACAM (purple circle not shown in the figure). The purple area is on the edge of land and is filtered out in Figure S24 and S26 (potentially due to clouds). Is it possible that undersampling issues in this area highlight the differences in spatial distributions in Figure 14? R square values in Figure S24-S26 are quite reasonable, considering that it is comparisons at fine-resolution. It is not convincing that the results in the manuscript suggest spatial allocation problems in NEI as written in the abstract.*

**Reply:**

Thank you for your thoughtful and valuable suggestions. We indeed didn't consider the sampling issue for the ACAM dataset. We have updated Figures 14, S24 – S27 and their captions (Figure S25 is new). Please see Lines 1299 – 1308 in the revised main

manuscript and Lines 179 – 200 in the revised supplemental figure file. Figure numbering has been updated accordingly. The updated figures have included the distributions of the relative differences of $NO_2$ VCDs between REAM and ACAM and the number of data points used to calculate grid cell mean $NO_2$ VCDs. The purple area in the old Figure 14 indeed has limited samplings ($\leq 3$); therefore, we have deleted the purple circle and relevant discussion about it. Please see Lines 687 – 690, 1299 – 1300 (Figure 14), and 1303 – 1304 (Figure 14 caption) in the revised main manuscript. However, these modifications do not affect our analysis results or conclusions, as we identified $NO_x$ emissions and gradients as a major factor causing the discrepancies among the 36-km REAM, the 4-km REAM, and observations through comprehensive evaluations and diagnostics of $NO_x$ related chemistry and physics (sections 3.6) and excluded the horizontal transport effect in Lines 682 – 687. The purple circle discussion is a detail that is not essential for the analysis.

The new Figure S25 shows the results only for those grid cells with samplings $\geq 10$. As expected, we can still find that the 4-km REAM has more concentrated $NO_2$ VCDs in Baltimore and Washington, D.C. urban regions but less concentrated in rural areas than ACAM in Figure S25. The following figure R2 shows the results without scaling $NO_2$ VCDs by corresponding domain averages. It considers the real differences of $NO_2$ VCDs between the 4-km REAM and ACAM. It is clear that the 4-km REAM generally has larger $NO_2$ VCDs in Baltimore and Washington, D.C. urban regions but lower $NO_2$ VCDs in rural areas than ACAM. Please see Lines 693 – 698 and 703 – 704 in the revised main manuscript.

$R^2$ or Pearson correlation coefficient is widely used in atmospheric science studies and our study. However, $R^2$ has limitations. The criteria of a good $R^2$ is arbitrary, depending on the purpose. Since NEI can identify urban regions using population densities, capturing the essential urban-rural contrast in $NO_x$ emissions, that $R^2$ value should be high. But that's not enough for our spatial distribution discussion, which is to say that $R^2$ does not describe how $NO_x$ emission decrease from urban to rural regions in the model as compared to the observations. For example, in Figure S27d in Line 199 in the revised supplemental figure file, we have a regression line *REAM = 2.11 ACAM – 1.11* with $R^2$ *= 0.61*. If we only focus on $R^2$, the regression is good. But this regression line generally shows larger REAM $NO_2$ VCDs than ACAM on high $NO_2$ VCD grid cells but lower REAM $NO_2$ VCDs than ACAM on low $NO_2$ VCD grid cells, as shown in Figure S27d and S27e. Even if Figure S27d shows a perfect regression line *REAM = 2.11 ACAM – 1.11* with $R^2 = 1$, the systematic biases between high $NO_2$ VCD grid cells and low $NO_2$ VCD grid cells persist. It is an obvious spatial distribution problem. In this situation, a perfect model-observation correlation requires $y = x$ with $R^2 = 1$. Therefore, we have added the distributions of the relative differences of scaled or unscaled $NO_2$ VCDs between REAM and ACAM in the updated figures to show the distribution issue directly. Besides comparing $NO_2$ VCDs between 4-km REAM and ACAM, we also showed other evidence for the NEI $NO_x$ emission distribution issue, including sitecomparison results and the comparison of NO$_2$ VCDs between the 36-km REAM and satellite products (section 3.7).

[Figure]

Figure R2. Same as Figure S25 in the revised supplemental figure file but without scaling NO$_2$ VCDs by corresponding domain averages in (a) and (b). The NO$_2$ VCD unit in (a) and (b) is $5 \times 10^{15}$ molecules cm$^{-2}$.

*References:*

*Compton, J. C., Delgado, R., Berkoff, T. A., and Hoff, R. M.: Determination of planetary boundary layer height on short spatial and temporal scales: A demonstration of the covariance wavelet transform in ground-based wind profiler and lidar measurements, Journal of Atmospheric and Oceanic Technology, 30, 1566-1575, https://doi.org/10.1175/JTECH-D-12-00116.1, 2013.*

*Steeneveld et al. (2007), Diagnostic equations for the stable boundary layer height: evaluation and dimensional analysis, Journal of Applied Meteorology and Climatology, 46, 212-225.*

*Koracin and Berkowics (1988), Nocturnal boundary-layer height: observations by acoustic sounders and predictions in terms of surface-layer parameters, Boundary-Layer Meteorology, 43, 65-83.*

*Nieuwstadt and Tennekes (1981), A rate equation for the nocturnal boundary-layer height, Journal of the Atmospheric Sciences, 38, 1418-1428.*

**References:**

Compton, J. C., Delgado, R., Berkoff, T. A., and Hoff, R. M.: Determination of planetary boundary layer height on short spatial and temporal scales: A demonstration of the covariance wavelet transform in ground-based wind profiler and lidar measurements, Journal of Atmospheric and Oceanic Technology, 30, 1566-1575, https://doi.org/10.1175/JTECH-D-12-00116.1, 2013.

Knepp, T. N., Szykman, J. J., Long, R., Duvall, R. M., Krug, J., Beaver, M., Cavender, K., Kronmiller, K., Wheeler, M., and Delgado, R.: Assessment of mixed-layer height estimation from single-wavelength ceilometer profiles, Atmos. Meas. Tech., 10, 3963-3983, https://doi.org/10.5194/amt-10-3963-2017, 2017.